# Trpm5 channels encode bistability of spinal motoneurons and ensure motor control of hindlimbs in mice

Rémi Bos [1,3✉], Benoît Drouillas[1,3], Mouloud Bouhadfane[1], Emilie Pecchi[1], Virginie Trouplin[1], Sergiy M. Korogod [2] & Frédéric Brocard [1✉]

Bistable motoneurons of the spinal cord exhibit warmth-activated plateau potential driven by $Na^+$ and triggered by a brief excitation. The thermoregulating molecular mechanisms of bistability and their role in motor functions remain unknown. Here, we identify thermosensitive $Na^+$-permeable Trpm5 channels as the main molecular players for bistability in mouse motoneurons. Pharmacological, genetic or computational inhibition of Trpm5 occlude bistable-related properties (slow afterdepolarization, windup, plateau potentials) and reduce spinal locomotor outputs while central pattern generators for locomotion operate normally. At cellular level, Trpm5 is activated by a ryanodine-mediated $Ca^{2+}$ release and turned off by $Ca^{2+}$ reuptake through the sarco/endoplasmic reticulum $Ca^{2+}$-ATPase (SERCA) pump. Mice in which Trpm5 is genetically silenced in most lumbar motoneurons develop hindlimb paresis and show difficulties in executing high-demanding locomotor tasks. Overall, by encoding bistability in motoneurons, Trpm5 appears indispensable for producing a postural tone in hindlimbs and amplifying the locomotor output.

---

[1] Institut de Neurosciences de la Timone (UMR7289), Aix-Marseille Université and CNRS, Marseille, France. [2] Bogomoletz Institute of Physiology, National Academy of Sciences of Ukraine, Kyiv, Ukraine. [3]These authors contributed equally: Rémi Bos, Benoît Drouillas. ✉email: remi.bos@univ-amu.fr; frederic.brocard@univ-amu.fr

The integrative function of the nervous system depends on the input–output properties of individual neural elements. The neural part of the motor unit, the motoneuron, integrates motor commands through a set of nonlinear properties to adjust muscle forces to behavioral needs[1–6]. One form of nonlinear input–output functions in motoneurons consists of a self-sustained firing evoked by a brief excitation and stopped by inhibition[7–11]. This bistable firing behavior is expressed in spinal motoneurons of mammals as early as birth[12], in adulthood under a monoaminergic control[13–16], and overexpressed after a spinal cord injury[17,18]. Bistability in motoneurons emerges from a slow afterdepolarization (sADP). When large enough, the sADP produces a sustained depolarization (plateau potential), which supports a self-sustained spiking[12]. Recordings of self-sustained spiking in motor units from awake animals[19–21] and humans[22–25] provide evidence that the plateau potential is part of the physiological repertoire of spinal motoneurons in mammals, although its motor functions remain uncertain[11,26,27].

The ionic mechanisms behind the sADP-related plateau potential have been investigated for decades. Among persistent inward currents, $Ca^{2+}$ entry through dendritic L-type $Ca^{2+}$ channels appears to play a prominent role in generating the plateau potential[28–32]. We also demonstrated that the activation of a $Ca^{2+}$-activated $Na^+$ conductance ($I_{CaN}$) secondary to the $Ca^{2+}$ entry drives the sADP-related plateau potential in lumbar motoneurons[12]. In addition to being $Ca^{2+}$ sensitive, the $Na^+$-permeant channel is warmth-activated making bistable motoneurons thermo-responsive[12]. Although the biophysical properties of channels mediating $I_{CaN}$ have been characterized, their molecular identity remains a mystery[33]. In the present study, we used a combination of cellular, electrophysiological, computational, behavioral, and genetic approaches to identify the channel(s) underlying $I_{CaN}$ in motoneurons and determine the functional role(s) of $I_{CaN}$-dependent plateau potentials in motor behaviors.

Our attention turned to a family of cationic channels called transient receptor potential (TRP) channels and in particular to two closely related TRP channels of the melastatin subfamily, Trpm4 and Trpm5. Both channels are (i) warmth-activated, (ii) $Ca^{2+}$-activated, (iii) permeable to $Na^+$ but not to $Ca^{2+}$, (iv) capable of maintaining a sustained depolarization[34–37]. Here we report Trpm5, but not Trpm4, as the main $Na^+$-permeant channel mediating the warmth-activated $I_{CaN}$ and provide evidence of its critical role in generating plateau potentials in bistable motoneurons. We also assigned to bistable motoneurons a behavioral role in both posture and high-demanding locomotor tasks.

## Results

**The sADP is predominantly mediated by a $Ca^{2+}$-activated $Na^+$ current ($I_{CaN}$).** The $I_{CaN}$ carried by $Na^+$ is critical for the sADP-related plateau potentials of lumbar motoneurons in rats[12]. We first ascertained that, in mice, large ventrolateral motoneurons recorded in L4–L5 (Fig. 1a) replicated features of the rat sADP after a brief depolarizing pulse. Namely, after attenuating voltage-gated $Na^+$ and $K^+$ channels with tetrodotoxin (TTX; 0.5–1 μM) and tetraethylammonium (TEA; 10 mM) respectively, a large part of the sADP: (i) emerged with $Ca^{2+}$ spikes (Fig. 1b), (ii) increased in amplitude with the number of $Ca^{2+}$ spikes (Fig. 1c), (iii) was stable over time (Fig. 1d), (iv) was strongly decreased by the-substitution of extracellular $Na^+$ (Fig. 1e) or chelation of intracellular $Ca^{2+}$ with BAPTA (10 mM; Fig. 1f), (v) was thermosensitive (Fig. 1g). Noteworthy, the sADP was more pronounced in mice relative to rats[12] and could ultimately give rise to a sustained regenerative depolarization (plateau potential;

Supplementary Fig. 1a). In normal artificial cerebrospinal fluid (aCSF; i.e., without TTX and TEA), most of large motoneurons (~90%) initially held near −60 mV displayed self-sustained spiking (the output of plateau potentials) in response to a brief supramaximal depolarization (Fig. 1h and Supplementary Fig. 1b). The bistable ability of each motoneuron was assessed by measuring the difference ($\Delta V$) between the most hyperpolarized ($V_h$ min) and the most depolarized ($V_h$ max) holding potentials at which self-sustained spiking can be triggered (Supplementary Fig. 1c). In bistable motoneurons, $\Delta V$ was of $7.3 \pm 0.5$ mV in mean, whereas it was null in non-bistable motoneurons (Supplementary Fig. 1d). As in rats[12], the proportion of bistable motoneurons decreased with the bath temperature (Fig. 1i). Also, the ability of the remaining bistable motoneurons to trigger a self-sustained spiking at low temperature decreased insofar as $\Delta V$ decreased to $2.8 \pm 1.0$ mV at 22–24 °C (Fig. 1j). Together, plateau potentials and the related self-sustained spiking are, to a large extent, dependent on a thermosensitive $Ca^{2+}$-activated inward current ($I_{CaN}$), with $Na^+$ as the primary charge carrier.

**The sADP mediated by $I_{CaN}$ is carried through Trpm5 channels.** Channels that give rise to $I_{CaN}$ in motoneurons were so far unidentified, though TRP cationic channels activated by $Ca^{2+}$ are likely candidates. Since sADP is thermosensitive, we focused on TRP channels gated by heat. Most of thermo-TRPs belong to Vanilloid (Trpv) and Melastatin (Trpm) subfamilies[38]. To evaluate their contribution, we used knock-out transgenic mice. Neither Trpv1-Trpv3 double knock-out nor Trpv2$^{-/-}$ mice displayed changes in the sADP amplitude (Supplementary Fig. 2a, b) or in the proportion of bistable motoneurons (Supplementary Fig. 2c, d). We then analyzed functional contributions of Trpm4 channels. The Trpm4 blocker 9-phenanthrol [50 μM,[39]] abolished self-sustained spiking in all bistable motoneurons recorded from wild-type mice (Supplementary Fig. 2e). Unexpectedly, the proportion of motoneurons to trigger self-sustained spiking and to generate a sADP was not altered in Trpm4$^{-/-}$ mice (Supplementary Fig. 2f, g). In fact, 9-phenanthrol was found to abolish $Ca^{2+}$ spikes and sADP concomitantly in control motoneurons (Supplementary Fig. 2h) and in Trpm4$^{-/-}$ motoneurons (data not shown). Therefore, at a concentration commonly used to block Trpm4 channels, 9-phenanthrol prevented bistable behaviors through a side-effect on voltage-gated $Ca^{2+}$ channels.

We next explored the functional significance of Trpm5 channels. We assessed their role by using the Trpm5 opener linoleic acid [L.A., 30–50 μM, ref. [40]] and the Trpm5 blocker triphenylphosphine oxide [TPPO, 30–50 μM, ref. [41]]. The amplitude of the sADP increased with L.A. (Fig. 2a), while TPPO had opposite effects (Fig. 2b). Noteworthy, the two drugs did not affect $Ca^{2+}$ spikes although input resistance was increased with TPPO (Supplementary Fig. 3a, b). The specificity of L.A. and TPPO to Trpm5 channels was confirmed by the lack of effects of the two drugs on both $Ca^{2+}$ spikes and the sADP recorded from Trpm5$^{-/-}$ mice (Supplementary Fig. 3c, d). Note that in Trpm5$^{-/-}$ mice, the sADP was smaller compared to wild-type animals (Fig. 2c, d) but similar to that of Trpm4-Trpm5 double knock-out mice (Supplementary Fig. 3e, f). To further explore the contribution of Trpm5 on the sADP, we performed its loss-of-function by using a short hairpin RNAs (shRNAs) against Trpm5. We first checked the efficiency of the Trpm5-shRNA to decrease the Trpm5 mRNA expression. In HEK-293 cells, Trpm5-shRNA reduced by ~95% the Trpm5 mRNA level (Fig. 2e). We then injected intrathecally at birth at T13–L1 level an adeno-associated virus (AAV9) encoding the Trpm5-shRNA with a green fluorescent reporter (eGFP). Near 12 days after the injection, the viral transfection led to a decrease of both mRNA and

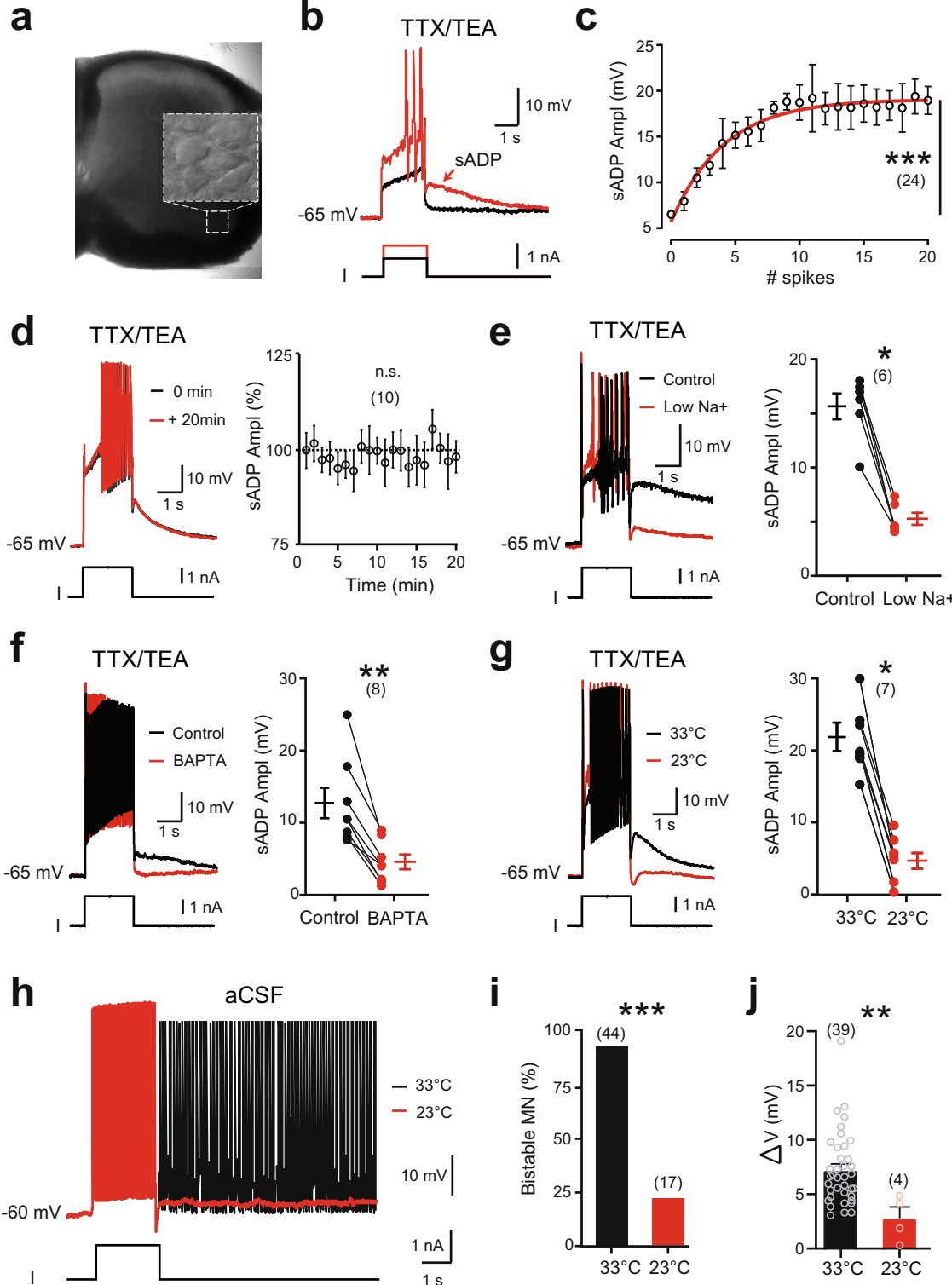

membrane protein expression of Trpm5 in the lumbar spinal cord by ~50 and ~15%, respectively (Fig. 2e, f). Concomitantly, a strong expression of eGFP in ventral horns of the spinal cord from T2–T3 to L5–S1 was observed (Fig. 2g) and 75 ± 3.6% of lumbar motoneurons (227 out of 299 large cholinergic neurons in the L4–L5 ventral horns from 4 mice) were transduced (arrows in Fig. 2h), whereas no transduced cells were seen in supraspinal structures (Supplementary Fig. 4). In the Trpm5-shRNA motoneurons, we observed a marked decrease of both the sADP

(Fig. 2i, ~50%) and the sADP-related tail inward current (Fig. 2j, ~60%) compared to motoneurons transduced with the scramble shRNA. We also noted that $19 ± 3.2\%$ of small ($<10\,\mu m$) non-cholinergic cells (122 out of 640) surrounded by many processes were also transduced by the AAV (double arrow in Fig. 2h). They were identified as astrocytes because they were $GFAP^+$ (not shown), not capable of firing action potentials, displayed a hyperpolarized resting membrane potential, and a low input resistance (Fig. 2k). Note that the electrophysiological properties

**Fig. 1 Functional characterization of the thermosensitive sADP in large lumbar motoneurons from mice. a** Acute spinal cord slice from the lumbar enlargement (L4) under infrared-differential interference contrast (IR-DIC) imaged at ×4 magnification. Inset: high magnification (×40) of a motoneuronal pool readily observable in the ventral horn and patch-clamped under IR-DIC. **b** Superimposition of voltage traces from a motoneuron recorded under TTX and TEA in response to subthreshold (black) or suprathreshold (red) depolarizing pulses. The arrow indicates the slow after depolarization (sADP). **c** Relationship between the peak amplitude of the sADP and the number of spikes emerging during a 2-s depolarizing current pulse. Continuous red line is the best-fit nonlinear regression ($n = 9$ mice). **d** Mean time-course changes in peak amplitude of the sADP. Values are relative to the amplitude of the first sADP ($n = 3$ mice). **e–g** Left: superimposition of voltage traces from motoneurons recorded under TTX and TEA, before and after removing $[Na^+]_o$ ($n = 3$ mice) (**e**), chelating intracellular $Ca^{2+}$ with BAPTA (10 mM) ($n = 2$ mice) (**f**), or decreasing temperature ($n = 3$ mice) (**g**), right: mean amplitude of the peak sADP. Each circle represents an individual motoneuron. **h** Superimposition of voltage traces recorded in normal aCSF (i.e., without TTX and TEA) in response to a 2-s depolarizing pulse before and after reducing the temperature of the bath from 33 to 23 °C. **i, j** Group mean quantification of the proportion of bistable motoneurons (**i**) and of the bistability range $\Delta V$ (**j**) as a function of temperature ($n = 13$ mice). Numbers in brackets indicate the numbers of recorded motoneurons. n.s., no significance; *$P < 0.05$; **$P < 0.01$; ***$P < 0.001$ (one-way ANOVA with multiple comparisons for **c, d**; two-tailed Wilcoxon paired test for **e–g**; two-tailed Fisher test for **i**; two-tailed Mann–Whitney test for **j**). Mean ± SEM. For detailed $P$ values, see Source data. Source data are provided as a Source data file. See also Supplementary Fig. 1.

of Trpm5-shRNA astrocytes were similar to those transduced with scramble shRNA (Fig. 2k). Altogether, we conclude that Trpm5 primarily mediates $I_{CAN}$ in spinal motoneurons.

**Trpm5 channels promote $I_{CaN}$-mediated bistability in motoneurons.** Next, we investigated the functional role of Trpm5 channels on the firing properties of bistable motoneurons by measuring their ability to generate self-sustained spiking. In normal aCSF, the activation of Trpm5 channels with L.A. enhanced the ability of bistable motoneurons to trigger self-sustained spiking as reflected by an increase in $\Delta V$ (Fig. 3a). The blockade of Trpm5 with TPPO had opposite effects by abolishing self-sustained spiking in most bistable motoneurons (Fig. 3b). Although passive membrane properties and the excitability of motoneurons recorded from $Trpm5^{-/-}$ mice were similar to wild-type animals (Supplementary Table 1), only half of them were bistable (Fig. 3c). Furthermore, the ability of these residual bistable motoneurons to trigger self-sustained spiking was weaker as reflected by a lower $\Delta V$ (Fig. 3c). Similar results were obtained from motoneurons transduced with Trpm5-shRNA with only ~30% of bistable motoneurons left (Fig. 3d and Supplementary Table 1). Importantly, bistable properties recorded from $Trpm5^{-/-}$ mice were not affected by L.A. or TPPO and were similar to those recorded from Trpm4-Trpm5 double knock-out mice (Supplementary Fig. 5 and Supplementary Table 1).

The other striking electrical manifestation of bistable motoneurons is the cumulative depolarization of their sADP induced by repetitive excitations (Fig. 3e–h). This apparent short-term memory, usually referred to as a "windup" phenomenon, corresponds to a partial activation of the plateau potential. In normal aCSF, the summation of sADPs was observed and increased by $5.7 \pm 0.5$ mV in response to 5 successive same-amplitude pulses. This windup phenomenon was, respectively, enhanced or attenuated when the Trpm5 opener L.A. or the Trpm5 blocker TPPO were bath applied (Fig. 3e, f). The windup was weaker in motoneurons transduced by the Trpm5-shRNA or recorded from $Trpm5^{-/-}$ mice (Fig. 3g, h).

**The recruitment of the Trpm5-mediated $I_{CaN}$ requires a $Ca^{2+}$-induced $Ca^{2+}$ release mechanism through ryanodine receptors (RyR).** We next investigated the mechanism responsible for Trpm5 activation. The sensitivity of the sADP to BAPTA may indicate a mobilization of the intracellular $Ca^{2+}$ stores. The $Ca^{2+}$ release could be mediated by: (i) the activation of a phospholipase C (PLC) signaling pathway leading to the formation of inositol (1.4.5)-triphosphate (IP3) and the subsequent activation of endoplasmic IP3 receptors, (ii) a direct gating of IP3 receptors by $Ca^{2+}$, and/or (iii) the activation of the RyR by $Ca^{2+}$. The PLC

inhibitor U73122 (10 μM) as well as the alkaloid IP3 receptor inhibitor xestospongin C (1–2.5 μM) did not affect the sADP (Fig. 4a, b). However, the inhibitor of RyR dantrolene (50 μM) reduced the sADP amplitude and the ability of bistable motoneurons to generate self-sustained spiking activity in normal aCSF (Fig. 4c, d). In contrast, the mobilization of $Ca^{2+}$ stores through the sensitization of RyR by caffeine (30 μM) increased the sADP, until ultimately a plateau depolarization developed (Fig. 4e). In addition, bistable motoneurons were more prone to triggering self-sustained spiking in normal aCSF when caffeine was applied (Fig. 4f).

Multiple cytoplasmic $Ca^{2+}$-sensing proteins detect the increase in free cytosolic $Ca^{2+}$ and operate in intracellular $Ca^{2+}$ signaling pathways to activate ion channels. Among them, protein kinase C (PKC) regulates many ion channels in a $Ca^{2+}$-dependent manner, including Trpm5 channels[42]. However, the broad-spectrum PKC blocker chelerythrin was ineffective in reducing the sADP (Fig. 4g). By contrast, the duration of the sADP was prolonged when the uptake of the cytosolic $Ca^{2+}$ by the sarco/endoplasmic reticulum $Ca^{2+}$-ATPase (SERCA) pump was inhibited by thapsigargin (Fig. 4h). Dantrolene, caffeine, and thapsigargin modulated sADP without affecting either $Ca^{2+}$ spikes or input resistance of the cell (Supplementary Fig. 6). Together, these results show that $Ca^{2+}$ entering through the voltage-gated $Ca^{2+}$ channels act as a trigger giving rise to further $Ca^{2+}$ release from intracellular stores through RyR and thereby recruits Trpm5 channels. Thereafter, by restoring $[Ca^{2+}]_i$ to resting levels, the SERCA pump allows for the slow decay of the sADP to the resting membrane potential.

**Trpm5 amplifies spinal locomotor outputs.** We next used our previous multi-compartment computational model[1] to investigate the effect of Trpm5 channels on motor output from a heterogeneous population of motoneurons. The model was supplemented with $I_{CaN}$ mediated by Trpm5-like channels for which the open probability increases steeply between 15 and 35 °C[35]. Individual simulated motoneurons expressing the Trpm5 current reproduced key features of the biological responses to depolarizing currents in the presence or absence of TTX/TEA [i.e., in reduced (Fig. 5a–c) or intact (Fig. 5d–f) $Na^+$ and $K^+$ conductances]. Specifically, the sADP and the self-sustained firing required a minimal set of holding potentials (Fig. 5a, d), temperature (Fig. 5b, e), and Trpm5 current (Fig. 5c, f). In sum, the model supplemented with Trpm5 current captures key features of both the sADP and self-sustained firing in bistable motoneurons, making it suitable as a tool to predict how Trpm5 channels might shape the motor output. We thus "cloned" the validated model to build a population of 50 uncoupled motoneurons with a randomized normal distribution of neuronal parameters including the

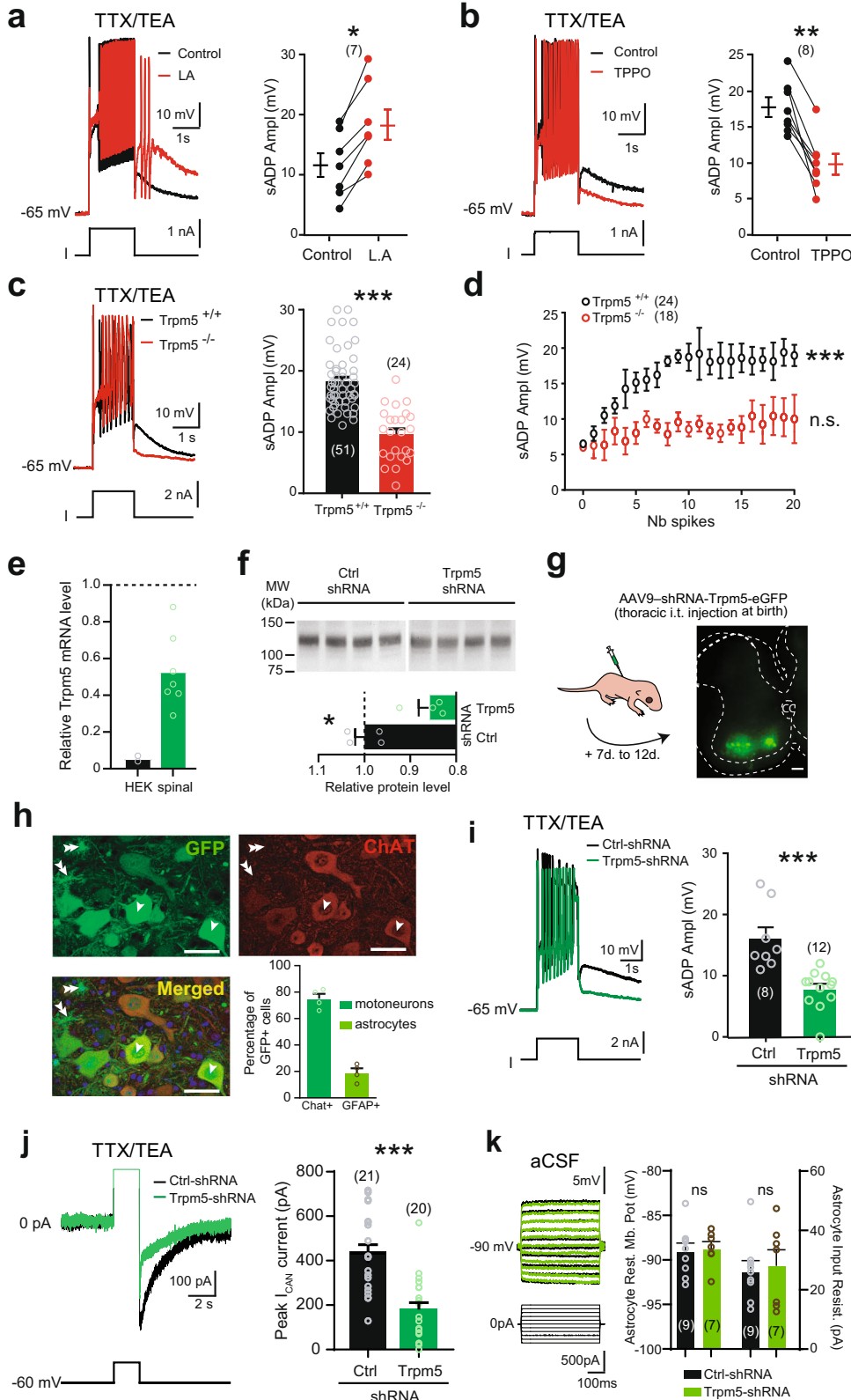

Trpm5 expression (see "Methods" and Table 1). Oscillatory synaptic excitation of either soma (Fig. 5g, j) or proximal dendrites (Fig. 5h, k) induced either a burst of spikes (red asterisk) on the top of each cycle or triggered a self-sustained firing, which started in an earlier (green asterisk) or later cycle (blue asterisk), depending on Trpm5 value. The number of in-burst spikes increased from cycle to cycle demonstrating a build-up of firing rate while the number of motoneurons generating self-sustained spiking progressively increased during the stimulus episode (Fig. 5g, h, upper panels). As a result, the integrated activity built up approaching a steady level (Fig. 5g, h middle and lower panels). When Trpm5 current was reduced to ~25% of its functionality, bistable motoneurons disappeared and the activity build-up decreased (Fig. 5i, l). Similar population effects were

**Fig. 2 The thermosensitive $I_{CaN}$-mediated sADP is driven by Trpm5 channels. a–c** Left: superimposition of voltage traces in motoneurons recorded under TTX/TEA from wild-type mice in response to a depolarizing pulse before and after bath-applying linoleic acid (**a**, L.A., 50 μM, $n = 3$ mice) or triphenylphosphine oxide (**b**, TPPO, 50 μM, $n = 4$ mice), or recorded in motoneurons from $Trpm5^{-/-}$ mice ($n = 5$ mice) (**c**), right: mean amplitude of the peak sADP. The numbers in brackets indicate the numbers of recorded motoneurons. Each circle represents an individual motoneuron. **d** Relationship between the peak amplitude of the sADP and the number of spikes emerging during a 2-s depolarizing current pulse in control (black, $n = 9$ mice) vs $Trpm5^{-/-}$ mice (red, $n = 5$ mice). **e** qRT-PCR analysis assessing the efficiency of the shRNA to knockdown Trpm5 mRNA in HEK-293 cell cultures ($n = 2$) and spinal cords ($n = 7$) from ~P12 mice. The expression level of the Trpm5 mRNA in cell cultures or spinal cords was normalized to scramble shRNA values with GAPDH or ACTB as internal references, respectively. Each circle represents the mean value from one cell culture or one spinal cord. **f** Up: Trpm5 immunoblots of lumbar segments from P12 mice intrathecally injected at birth with an adeno-associated virus (AAV9) encoding either a scramble shRNA ($n = 4$ mice) or a Trpm5-targeting shRNA ($n = 4$ mice). One mice per lane. Bottom: group mean quantification of the ~130 kDa band normalized to scramble-injected controls. **g** Left: schematic representation of the experimental design, right: acquisition of a transverse spinal slice (L4) from a P10 mouse intrathecally injected at birth with an AAV9 encoding Trpm5-targeting shRNA and eGFP. Scale bar = 100 μm. The experiment was repeated four independent times with similar results. **h** High magnification of the ventral horn showing native fluorescence of motoneurons (single arrow) transduced by AAV9 (upper left) and immunostained for choline acetyltransferase (upper right, ChAT antibody; bottom left, merged images). Some astrocytes (double arrow) were also GFP+. Histograms (bottom right): group mean quantification of the proportion of 299 motoneurons (green) and 640 astrocytes (orange) from 4 mice transfected by AAV9-shRNA-Trpm5-eGFP. Each circle represents one mouse. Scale bar = 50 μm. **i, j** Left: superimposition of voltage (**i**) or current (**j**) traces from GFP+ motoneurons recorded under TTX/TEA and transduced either with scramble shRNA (black, $n = 6$ mice) or with a Trpm5-targeting shRNA (green, $n = 5$ mice), right: mean amplitude of the peak sADP (**i**) and the peak amplitude of the $I_{CAN}$ current (**j**). **k** Left: superimposition of voltage traces from GFP+ astrocytes recorded in normal aCSF and transduced either with scramble shRNA (black, $n = 3$ mice) or with a Trpm5-targeting shRNA (green, $n = 3$ mice), right: mean amplitude of the astrocytic resting membrane potential (left) and the input resistance (right). The numbers in brackets indicate the numbers of recorded cells. Each circle represents an individual motoneuron or astrocyte. *$P < 0.05$; **$P < 0.01$; ***$P < 0.001$ (two-tailed Wilcoxon paired test for **a**, **b**; two-tailed Mann–Whitney test for **c**, **f**, **i–k**; one-way ANOVA with multiple comparisons for **d**). Mean ± SEM. For detailed $P$ values, see Source data. Source data are provided as a Source data file. See also Supplementary Figs. 2–4.

observed when motoneurons were stimulated by depolarizing current pulses (1 Hz, 200 ms; not illustrated) that evidenced robustness of the build-up phenomena and their Trpm5-dependence.

To confirm the predictive functional role of Trpm5 channels in building up motor outputs, we performed ex vivo experiments from whole-mount spinal cord preparations (Fig. 6a). In response to sensory inputs (evoked by brief repetitive dorsal root stimuli), a typical sustained ventral root discharge developed and increased in amplitude in a windup manner (Fig. 6b–e). The genetic deletion of Trpm5 channels reduced the motoneuron spiking probability and occluded the windup discharge (Fig. 6b, d). Similar results were obtained when the Trpm5 blocker TPPO was bath applied in spinal cords from wild-type animals (Fig. 6c, e). Note that the monosynaptic response was not affected by TPPO suggesting that the glutamatergic synaptic transmission was not compromised by the drug (Supplementary Fig. 7a).

The recurrent partial activation of plateau potentials in motoneurons has also been suggested to increase the amplitude of the locomotor drive from the CPG[4]. To test the role of motoneuron Trpm5 channels in the integration of rhythmic locomotor inputs, we evoked fictive locomotor activities by $N$-methyl-DL-aspartic acid (NMA)/5-hydroxytryptamine (5-HT) (see "Methods"). During fictive locomotion, the application of TPPO decreased the locomotor burst amplitude without affecting either the locomotor pattern (left–right, flexor–extensor alternations), cycle period, or burst duration (Supplementary Fig. 7b). The reduction of the locomotor burst amplitude could reflect a decreased activity of lumbar motoneurons and possibly of locomotor-related premotor neurons. To avoid possible effects on rhythm-generating networks mainly located at L2, drugs targeting Trpm5 channels were applied to motoneuron pools caudally located (Fig. 6f). A similar decrease of locomotor outputs in the L5 compartment superfused with TPPO was observed suggesting that Trpm5 channels are functionally activated in motoneurons during locomotor circuit activity in vitro. Note that the spinal cords isolated from neonatal $Trpm5^{-/-}$ mice exhibited a quite normal alternating locomotor pattern (Supplementary Fig. 7c). Together, these data indicate that Trpm5 channels are dispensable for the locomotor rhythm generation and

coordination, but their activation at the level of motoneurons amplifies locomotor outputs.

**Role of Trpm5 in motor behaviors**. Adult $Trpm5^{-/-}$ mice appeared healthy with grossly normal home cage behavior but neonatal $Trpm5^{-/-}$ mice displayed a weaker ability to flip from a supine position (Fig. 7a and Supplementary Movie 1). As soon as mice were able to walk by themselves at ~12 days of age, $Trpm5^{-/-}$ mice walked with a regular and typical alternating locomotor pattern but with a wider base of support (Fig. 7b). Similar locomotor phenotype was observed in young (3 weeks) adult Trpm4-Trpm5 double knock-out mice (Supplementary Fig. 8). In more challenging locomotor tasks, $Trpm5^{-/-}$ mice (Trpm4-Trpm5 double knock-out mice were not tested) failed to adapt to accelerated speed in the rotarod test (Fig. 7c, d) and displayed a lower swimming performance (Fig. 7e–g and Supplementary Movie 2). However, all locomotor deficits faded slowly over weeks and became inconspicuous after 5 weeks of age in comparison to wild-type littermates (Fig. 7a–f).

To specifically discriminate the behavioral role of Trpm5 in lumbar motoneurons, we transduced them by means of a lumbar intrathecal injection of the AAV9-Trpm5-shRNA, as shown above (Fig. 2g). Mice transduced with Trpm5-shRNA showed striking motor deficits compared to control-shRNA mice (Fig. 7h). They required more time to flip to a supine position without noticeable motor improvements with age (Fig. 7i and Supplementary Movie 4). At the second postnatal week, Trpm5-shRNA animals did not exhibit postural tone in hindlimbs and typically dragged themselves around the cage using the forelimbs (Fig. 7h and Supplementary Movies 3 and 4). They also displayed lower swim performance insofar as they mainly propelled by means of forelimbs (Fig. 7j and Supplementary Movie 5). At the end of the third postnatal week, due to the total lack of hindlimb postural tone, mice could not be tested on rotarod or swim tank. Mice transduced with the scramble shRNA displayed a delay in the development of the ability to flip to a supine position (Fig. 7i) compared to not injected wild-type animals (Fig. 7a). Nonetheless, control-shRNA mice acquired normal righting reflex and a quadrupedal stance with body weight support at the end of the

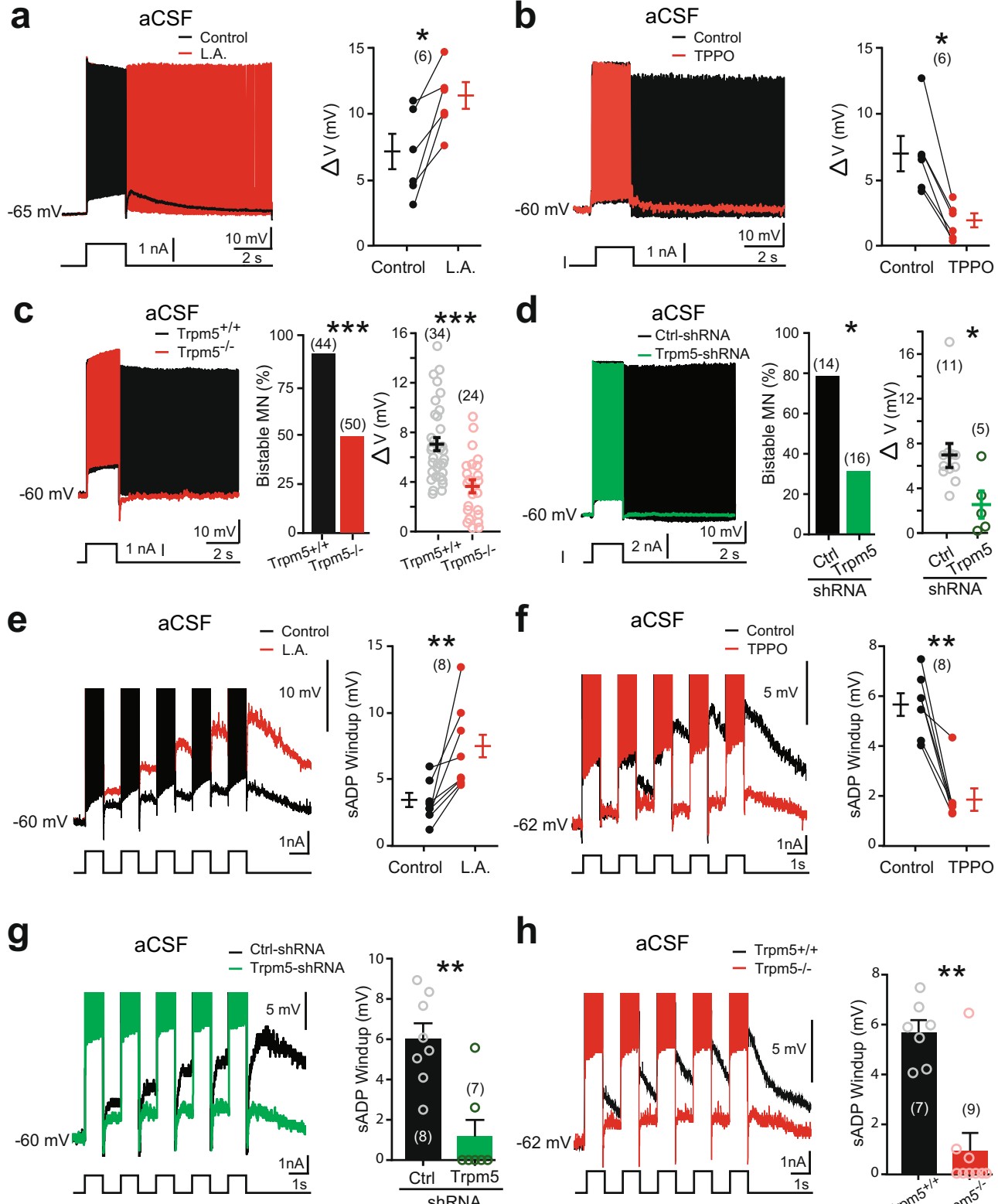

**Fig. 3 Bistability of motoneurons relies on Trpm5 channels. a–h** Left: superimposition of voltage traces recorded in motoneurons from wild-type mice in response to a single (**a–d**) or repetitive (1 Hz, **e–h**) depolarizing current pulses before (black) and after (red) bath-applying linoleic acid (L.A., 30 μM) (**a**, **e**, $n = 5$ mice), or triphenylphosphine oxide (TPPO, 30 μM, $n = 7$ mice) (**b**, **f**), or recorded in motoneurons from Trpm5$^{-/-}$ mice (**c**, **h**, $n = 7$ mice) or from eGFP+ motoneurons transduced either with the scramble (black, $n = 5$ mice) or with a Trpm5-targeting shRNA (green, $n = 7$ mice) (**d**, **g**), right: group mean quantification of the proportion of bistable motoneurons and/or $\Delta V$ (**a–d**) and of the sADP windup (**e–h**). Numbers in brackets indicate the numbers of recorded motoneurons. Each circle represents an individual motoneuron. *$P < 0.05$; **$P < 0.01$; ***$P < 0.001$ (two-tailed Wilcoxon paired test for **a**, **b**, **e**, **f**; two-tailed Fisher test for **c**, **d** (middle histograms); two-tailed Mann–Whitney test for **c** (right), **d** (right), **g**, **h**). Mean ± SEM. For detailed $P$ values, see Source data. Source data are provided as a Source data file. See also Supplementary Fig. 5.

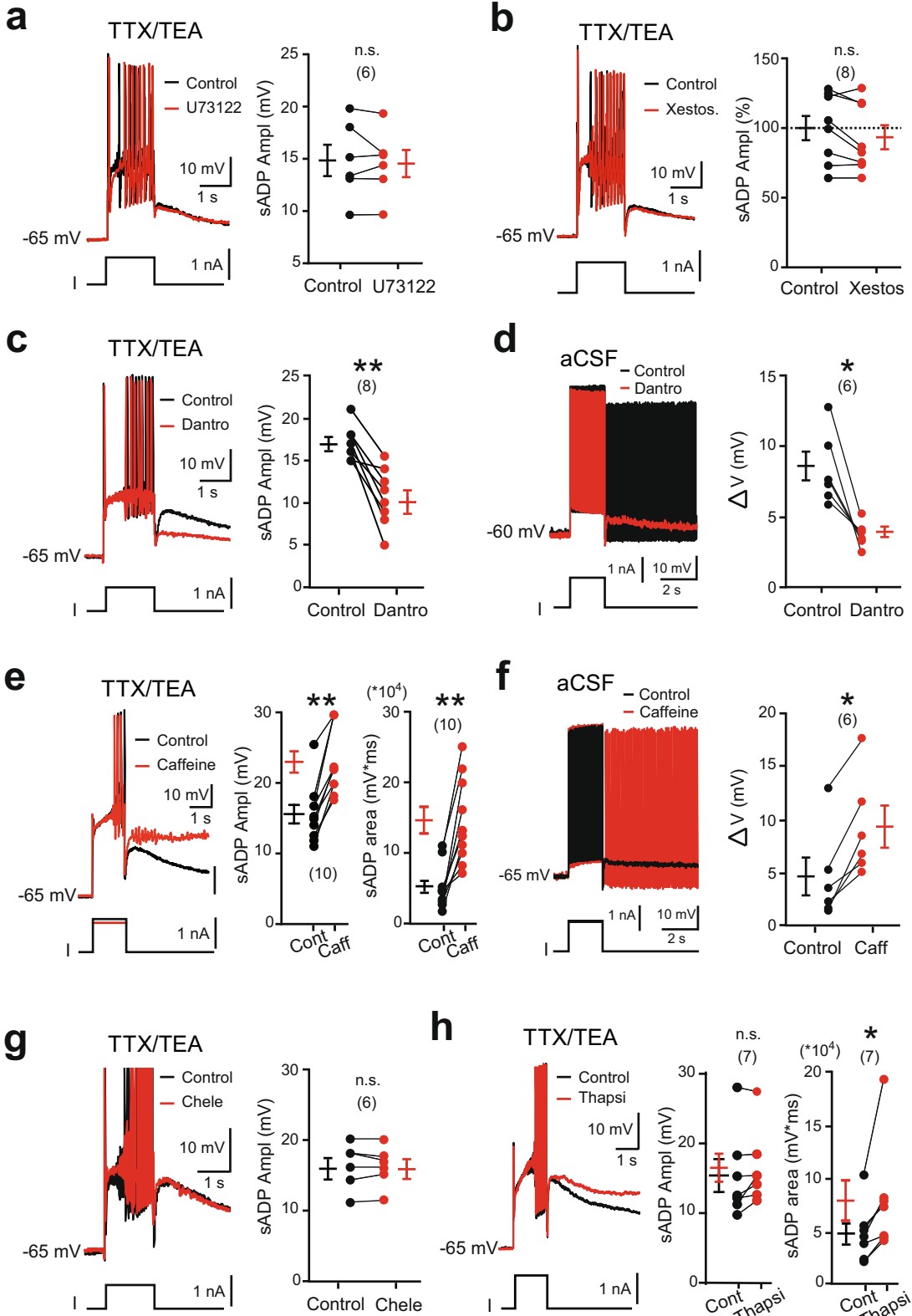

second postnatal week (Fig. 7h, i and Supplementary Movies 3 and 4). In sum, Trpm5-mediated $I_{CAN}$ in lumbar motoneurons appears to play a key role for the motor control of hindlimbs.

## Discussion

The present study provides insights into the operation of the motor network: (i) it identifies Trpm5 as the main channel

mediating $I_{CAN}$ and driving the sADP-related plateau potential in spinal motoneurons, (ii) it shows the critical role of ryanodine-sensitive $Ca^{2+}$ store and SERCA pumps in recruiting Trpm5 and slowing down the sADP, respectively, (iii) it provides a paradigm shift in ionic mechanisms underlying bistability by considering L-type $Ca^{2+}$ channels as the trigger rather than the charge provider of plateau potentials, (iv) it assigns to bistable motoneurons

**Fig. 4 Ryanodine-operated Ca$^{2+}$ release activates Trpm5 to promote bistability in motoneurons. a–h** Left: superimposition of voltage traces from motoneurons in response to a 2-s depolarizing current pulse recorded with (**a–c**, **e**, **g**, **h**) or without (**d**, **f**) TTX/TEA before and after bath-applying U73122 (**a**, 10 μM, $n = 2$ mice), xestospongin C (**b**, 1–2.5 μM, $n = 2$ mice), dantrolene (**c**, **d**, 30 μM, $n = 5$ mice), caffeine (**e**, **f**, 30 μM–5 mM, $n = 5$ mice), chelerythrin (**g**, 10 μM, $n = 3$ mice), or thapsigargin (**h**, 1 μM, $n = 3$ mice), right: quantification of the area and/or amplitude of the sADP (**a–c**, **e**, **g**, **h**) or of the $\Delta V$ (**d**, **f**) defined as the difference between the most depolarized pre-stimulus holding potential and the most hyperpolarized holding potential for which self-sustained firing can be triggered (see Supplementary Fig. 1). Numbers in brackets indicate the numbers of recorded motoneurons. n.s., no significance; *$P < 0.05$; **$P < 0.01$ (two-tailed Wilcoxon paired test). Mean ± SEM. For detailed $P$ values, see Source data. Source data are provided as a Source data file. See also Supplementary Fig. 6.

a role in motor behaviors related to posture and high-demanding locomotor tasks.

In line with our previous study performed in rats[12], several of our experimental data conclude that sADP in mice is mainly generated by $I_{CaN}$ carried by Na$^+$. We initially attributed $I_{CaN}$ to Trpv2 channels in spinal motoneurons[12]. However, Trpv2$^{-/-}$ mice failed to confirm contribution of Trpv2 channels to sADP (Supplementary Fig. 2b). It was recently concluded that Trpm4 channels promote spontaneous bursting activities in the spinal cord[43]. These channels are expressed in spinal motoneurons[44], but Trpm4$^{-/-}$ mice did not display alteration of the sADP (Supplementary Fig. 2g). Note that the effect of 9-phenantrol on sADP (Supplementary Fig. 2h) did not result from the blockade of Trpm4 since the sADP in Trpm4$^{-/-}$ mice was equally sensitive to 9-phenantrol. Thus, the simultaneous blockade of the sADP and Ca$^{2+}$ spikes by 9-phenantrol suggests a side effect on voltage-gated Ca$^{2+}$ currents, as originally described on L-type Ca$^{2+}$ channels[45]. In support, blockade of L-type Ca$^{2+}$ channels replicates the effect of 9-phenantrol in spinal motoneurons[12]. Rather than Trpm4, we identify Trpm5 channel as the major contributor to sADP. First, the sADP was strongly reduced after deleting, silencing or inhibiting Trpm5 (Fig. 2). Second, a computational model endowed with a Trpm5-like conductance reproduced motoneuron bistable properties, occurrence and disappearance of sADP under the same conditions as in biological prototype (Fig. 5). Third, in situ hybridization revealed Trpm5 expression in mice lumbar motoneurons (© 2008 Allen Developing Mouse Brain Atlas. Available from: https://developingmouse.brain-map.org). Fourth, the sADP has substantial similarities with the biophysical profile of the Trpm5-encoded current in heterologous expression systems[35,36,46,47].

As a residual sADP is still observed regardless of the method used to inhibit Trpm5 function, the sADP must depend on supplementary channel(s). TRP channels belonging to the canonical family (*Trpc*) are attractive. They participate to $I_{CaN}$-dependent sADP in the pre-Bötzinger complex[48–50], the cortex[51], or in the corticolimbic system[52]. In addition, the inactivation of Kv1.2 channels responsible for a slow ramping depolarization in motoneurons may also be responsible of the sADP[1]. Further studies will be required to identify channel(s) working in conjunction with Trpm5 to generate the residual sADP.

What are the Ca$^{2+}$ sources of Trpm5 activation? From chemosensory transduction studies, the current model involves a PLC signaling cascade that opens Trpm5 channels via an IP3-mediated Ca$^{2+}$ release[47,53]. We showed in motoneurons that bradykinin recruits a Na$^+$-mediated $I_{CaN}$ via a similar signaling pathway[54]. A possibility is that neuromodulators activating a G protein-coupled receptors recruit Trpm5 channels. In our study, the insensitivity of the sADP to U73122 or xestospongin shows that $I_{CaN}$ activated by a brief depolarization does not depend on a PLC signaling cascade. However, the activity of G protein-coupled receptors is expected to be minimal in slice preparations with no neuromodulators added. The finding that Ca$^{2+}$ spikes are closely associated with the induction of the sADP raises the possibility that Ca$^{2+}$ influx from the extracellular milieu directly

activates Trpm5 channels. However, this hypothesis is unlikely because macroscopic L-type Ca$^{2+}$ currents can be evoked in BAPTA-containing motoneurons without triggering the sADP [Fig. 1f; ref. [12]]. Finally, manipulations of RyR by dantrolene and caffeine show that the Ca$^{2+}$ source for activating Trpm5 channels mainly derives from internal stores.

In sum, as depicted in Fig. 8, it appears that Ca$^{2+}$ entry, at least via L-type Ca$^{2+}$ channels, triggers a Ca$^{2+}$-induced Ca$^{2+}$-release mechanism that subsequently activates Trpm5 channels leading to the sADP due to the influx of Na$^+$. Once the sADP reaches the spiking threshold, a self-enforcing process takes place for generating a long-lasting plateau potential (Fig. 8). The facilitation of self-sustained spiking by caffeine in the human neuromuscular system agrees with this cascade of events[55]. In this scenario, contrary to what has been thought for decades, L-type Ca$^{2+}$ channels do not provide long-lasting currents that lead to plateau potentials. In line with this, low-threshold L-type Ca$^{2+}$ channels can be supplanted by the high-threshold voltage Ca$^{2+}$ channels for generating the plateau potential[12]. Although dispensable, L-type Ca$^{2+}$ channels are very efficient for fueling motoneurons in Ca$^{2+}$ required to activate $I_{CaN}$ and their high expression on the soma and proximal dendrites in motoneurons[56] may mediate Ca$^{2+}$ entry in response to summed excitatory input to trigger the Trpm5-mediated plateau potential, as observed in our model.

There is so far no clear demonstration for the involvement of $I_{CaN}$ in locomotor rhythm and pattern generation[57]. In line with this, Trpm5$^{-/-}$ mice display normal step cycle and coordination regardless of age. Furthermore, Trpm5 channels are dispensable for the expression of intrinsic rhythmogenic activity in isolated spinal cord preparations. In the respiratory circuits, $I_{CaN}$ generates intrinsic bursting activities[58,59] and amplifies motor outputs rather than playing a fundamental role in the rhythm generation[49,50]. We reach a similar conclusion within the spinal locomotor network. The fictive locomotor output was reduced when Trpm5 was selectively abolished over the caudal-most lumbar motoneurons. We thus assume that the activation of voltage-gated Ca$^{2+}$ channels recruits Trpm5 in motoneurons on a cycle-to-cycle basis to amplify locomotor outputs. Such mechanism might be critical for high-demanding locomotor tasks. Indeed, without Trpm5 locomotor performances are impaired in walking at high speed or swimming; especially when fur-related buoyancy is less important in young animals. As the number of spikes increases when the locomotion shifts to a higher speed, it is conceivable that the recruitment of Trpm5 channels increases to progressively amplify the locomotor drive to produce a powerful motor output. In support, plateau-like potentials in cat motoneurons increases at depolarized levels during locomotion and has been suggested to amplify the locomotor-related excitation[4].

Several studies have speculated on the roles of bistable motoneurons and their plateau potentials in motor functions[3,26,27]. The present study provides evidence that Trpm5 channels and the related plateau potentials in bistable motoneurons are determinant for producing a postural tone. Silencing Trpm5 in lumbar motoneurons by shRNA leads to a pronounced paresis of

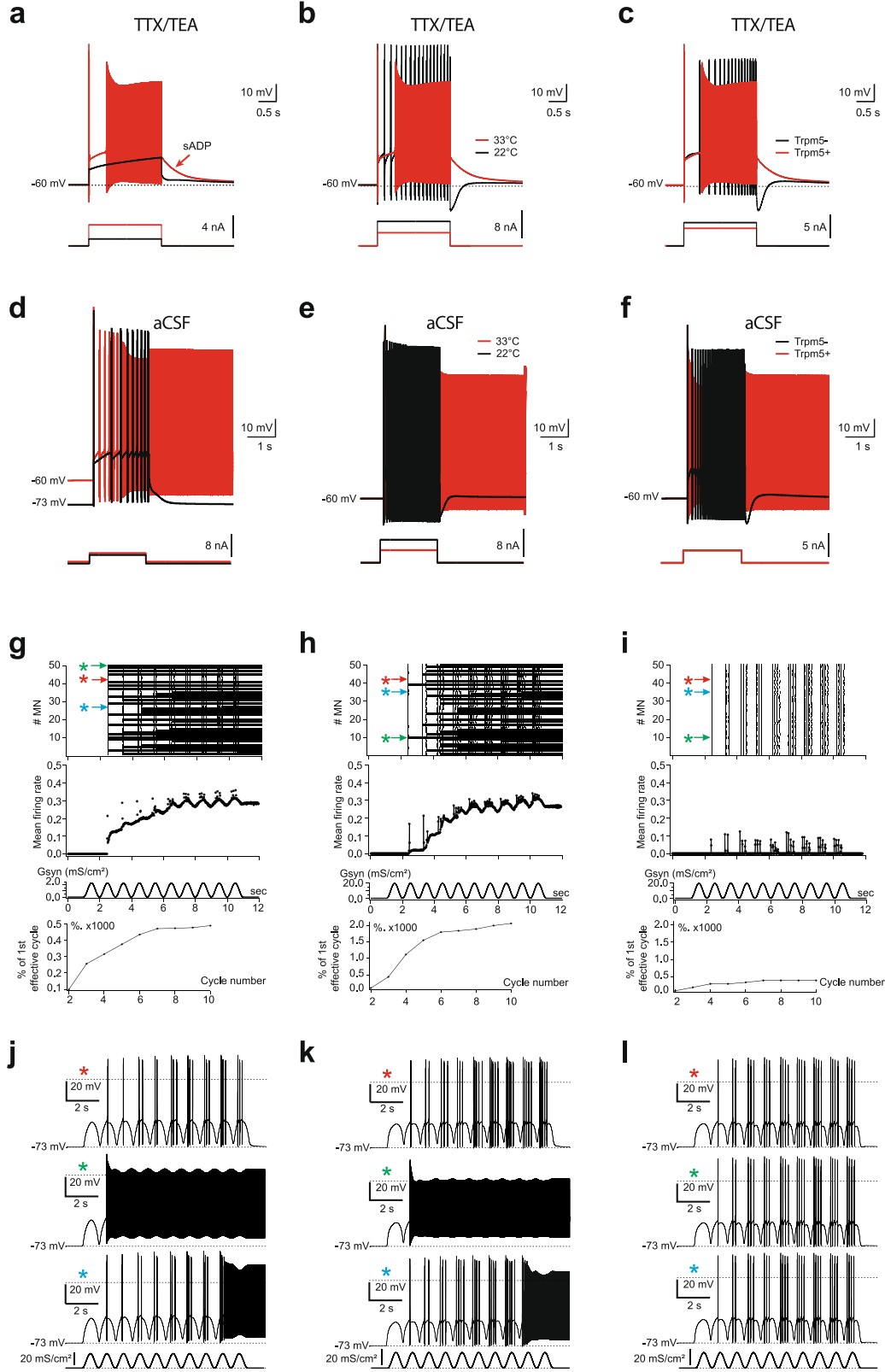

hindlimbs suggesting that the self-sustained discharge is a cellular correlate of the postural tone. Functionally, the increase of body temperature from rodents during the first two postnatal weeks[60] may contribute to the gradual acquisition of the quadrupedal stance subsequent to the emergence of thermosensitive Trpm5-mediated plateau potentials in motoneurons (Fig. 1h–j). The improvement of motor performances in Trpm5[−/−] mice may

suggest a lesser role of Trpm5 channels in adults and/or a developmental compensatory mechanism which is usual in knock-out mice. Further study will be required in the future to specifically investigate the role of Trpm5 in bistable properties of motoneurons in adults. We noted that the behavioral consequence of Trpm5 silencing on postural tone is unlikely related to a predominant astrocytic component. Indeed, astrocytes are

**Fig. 5 Simulated motoneurons supplemented with Trpm5 channels display self-sustained spiking activity and predict a role of Trpm5 in amplifying motor outputs. a–c** Superpositions of voltages generated by simulated motoneuron with diminished conductance of $Na^+$ and $K^+$ channels (TTX/TEA condition) in response to depolarizing 2-s pulses (bottom insets) applied at the soma initially held at −60 mV. The sADP (arrow in **a**) followed the spiking evoked by suprathreshold stimuli (red) in case of warm temperature (33 °C) and intact Trpm5 channels (Trpm5+) but disappeared (black) after reducing stimulation to subthreshold values (**a**), cooling to 22 °C (**b**), or blockade of Trpm5 channels (**c**). **d–f** Self-sustained spiking activity (red) of simulated motoneuron with intact $Na^+$ and $K^+$ channels triggered by a 2-s depolarizing stimuli in case of pre-holding at −60 mV, warm temperature of 33 °C, and intact Trpm5 channels (Trpm5+). Self-sustained spiking did not occur if Trpm5 channels were not sufficiently activated due to low holding (background) potential of −73 mV (**d**), temperature decrease to 22 °C (**e**), or the Trpm5 channels were totally blocked (**f**) although the cell remained capable of firing in response to a depolarizing pulse (black traces). **g–i** Integrated firing activity in a population of 50 uncoupled motoneurons with randomized Trpm5 expression (normal distribution of maximum conductivity $G_{Trpm5}$, mean ± s.d. = 55 ± 11 mS/cm$^2$) generated in response to 1-Hz sinusoid synaptic excitation of the soma (**g**) or dendrites (**h**). **i** same as in (**h**), but for an ~4-fold reduced Trpm5 expression ($G_{Trpm5}$ mean ± s.d. = 12.75 ± 2.0 mS/cm$^2$). Panels top to bottom: raster plots of spiking; mean firing rate in spikes per 1 s per neuron; synaptic conductivity $G_{syn}$ associated with 0-mV reversal potential; normalized cycle-to-cycle firing rate in percentage of response to first effective cycle. Arrows in the top panel indicate scatter plots of firing of individual neurons, exemplified below in **j–l** and marked by asterisks of the corresponding color.

**Table 1 Partial maximum conductivities (S/cm$^2$).**

|  | $G_{TRPM5}$ | $G_{NaTTX}$ | $G_{NaP}$ | $G_{CaN}$ | $G_{CaL}$ | $G_{fKDR}$ | $G_{Kv1.2}$ | $G_{KSK}$ | $G_{Leak}$ |
|---|---|---|---|---|---|---|---|---|---|
| Soma | 0.045, varied | 0.5 | 0.0008 | 0.013 | 0.05 | 0.36 | 0.01 | 0.01 | 5.3e−5 |
| Axon hillock | — | 0.5 | — | — | — | 0.36 | 0.01 | — | 5.3e−5 |
| Initial segment | — | 0.5 | — | — | — | 0.36 | 0.01 | — | 5.3e−5 |
| Nodes of Ranvier | — | 0.5 | — | — | — | 0.36 | — | — | 2.5e−5 |
| Myelinated segments | — | — | — | — | — | — | — | — | 2.5e−5 |
| Dendrites | — | — | — | — | — | — | — | — | 5.3e−5 |

much less transduced by the AAV9-shRNA-Trpm5 than motoneurons and their electrophysiological properties are not altered. This result is in line with the lack expression of Trpm5 in astrocytes as reported in the optic nerve[61].

Overall, the study provides biological insights in bistability of spinal motoneurons by demonstrating the significant role of $Ca^{2+}$-activated $Na^+$-permeable Trpm5 channels, and how these channels work in tandem with L-type $Ca^{2+}$ channels to set the gain of motor outputs. It also brings a clear support to the concept that the long-lasting maintenance of postural tone during standing is associated with bistable properties of motoneurons at least in young animals, while powerful motor output during locomotion is likely associated with cyclic activation of Trpm5 channels.

## Methods

**Experimental model**. Mice (C57/Bl6 background) of either sex (P1–P3 for fictive locomotion experiments, P5–P6 for windup experiments from whole-mount spinal cord, P5–P12 for patch-clamp recordings, P5–P12 for swimming tests, P1–P12 for surface righting reflex experiments, P9–P12 for base of support tests) and young adult (3, 4, and 5 weeks old for Catwalk and base of support tests, 3–4 weeks for swimming experiments, 4–5 weeks for rotarod experiments) were housed under a 12 h light/dark cycle in a with ad libitum access to water and food. Room temperature was kept between 21 and 24 °C and between 40 and 60% relative humidity. All transgenic mice were generated from the same C57/Bl6 genetic background. Trpv1-3$^{-/-}$, Trpv2$^{-/-}$, Trpm4$^{-/-}$, and Trpm5$^{-/-}$ mice were generated by and obtained from Aziz Moqrich[62], Michael Caterina[63], Pierre Launay[64], and Robert F. Margolskee[65], respectively. Double knock-out mice were generated by intercrossing Trpm4$^{-/-}$ and Trpm5$^{-/-}$ mice. All animal care and use were conformed to the French regulations (Décret 2010-118) and approved by the local ethics committee (Comité d'Ethique en Neurosciences INT-Marseille, CE71 Nb A1301404, authorization Nb 2018110819197361).

**shRNA constructs**. Specific shRNA sequence designed to knockdown Trpm5 transcript (GCTCGTGTGAACTGTTCTCTT) was incorporated into an adeno-associated viral (AAV) vector (serotype 9), which features a U6 polymerase promoter to drive shRNA expression and a CMV promoter to drive eGFP expression for identification of transduced neurons (Vector Builder, Chicago, IL). We also used a non-targeting shRNA sequence (CCTAAGGTTAAGTCGCCCTCG), which has no homology to any known genes in mouse as a control. The standard titers of AAVs were ≥1 × 10$^{13}$ GC/ml (genome copies/ml).

**Intrathecal vector delivery**. A minimally invasive technique was used to micro-inject AAV vectors into the T13–L1 intervertebral space. Briefly, in pups cryoanesthetized at birth, the intervertebral space was widened by flexing the spine slightly. The tip of the microcapillary preloaded with the AAV particles was lowered into the center of the T13–L1 intervertebral space. A total volume of 2 μl/animal was then slowly injected by hand.

**In vitro preparations**

*Slice preparation*. For the slice preparation, the lumbar spinal cord was isolated in ice-cold (+4 °C) aCSF solution composed of the following (in mM): 252 sucrose, 3 KCl, 1.25 KH$_2$PO$_4$, 4 MgSO$_4$, 0.2 CaCl$_2$, 26 NaHCO$_3$, 25 D-glucose, pH 7.4. The lumbar spinal cord was then introduced into a 1% agar solution, quickly cooled, mounted in a vibrating microtome (Leica, VT1000S) and sliced (325 μm) through the L4–5 lumbar segments. Slices were immediately transferred into the holding chamber filled with bubbled (95% $O_2$ and 5% $CO_2$) aCSF solution composed of (in mM): 120 NaCl, 3 KCl, 1.25 NaH$_2$PO$_4$, 1.3 MgSO$_4$, 1.2 CaCl$_2$, 25 NaHCO$_3$, 20 D-glucose, pH 7.4, 30-32 °C. After a 30–60 min resting period, individual slices were transferred to a recording chamber continuously perfused with aCSF heated to 32–34 °C.

*Whole-spinal cord preparation*. For the whole-spinal cord preparation, the spinal cord was transected at T8–9, isolated and transferred with intact dorsal and ventral roots to the recording chamber. The tissue was continuously bubbled (95% $O_2$ and 5% $CO_2$) and perfused with heated (~27–28 °C) aCSF solution composed of (in mM): 120 NaCl, 4 KCl, 1.25 NaH$_2$PO$_4$, 1.3 MgSO$_4$, 1.2 CaCl$_2$, 25 NaHCO$_3$, 20 D-glucose, pH 7.4.

**In vitro recordings and stimulations**

*Slice preparation*. For the slice preparation, whole-cell patch-clamp recordings were performed using a Multiclamp 700B amplifier (Molecular Devices) from L4–L5 motoneurons with the largest soma (>400 μm$^2$) located in the lateral ventral horn. Patch electrodes (2–4 MΩ) were pulled from borosilicate glass capillaries (1.5 mm OD, 1.12 mm ID; World Precision Instruments) on a Sutter P-97 puller (Sutter Instruments Company) and filled with an intracellular solution (in mM): 140 K$^+$-gluconate, 5 NaCl, 2 MgCl$_2$, 10 HEPES, 0.5 EGTA, 2 ATP, 0.4 GTP, pH 7.3. In some recordings, 10 mM of BAPTA or 1–2.5 μM of Xestospongin-C was added in the pipette solution to chelate intracellular free $Ca^{2+}$ or to inhibit InsP3 receptors from the intracellular stores. For astrocyte recordings, pipettes (7–9 MΩ) were filled with an intracellular solution composed of (in mM): 105 K$^+$-gluconate, 10 NaCl, 20 KCl, 0.15 MgCl$_2$, 10 HEPES, 0.5 EGTA, 4 ATP, 0.3 GTP, pH 7.3. GFP positive astrocytes (control ShRNA vs Trpm5-ShRNA) were identified on the basis of their small size (~10 μm) and characteristic morphology of a round soma surrounded by many processes. Pipette and neuronal capacitive currents were canceled and, after breakthrough, the series resistance was compensated and monitored. Recordings were digitized on-line and filtered at 10 kHz through a

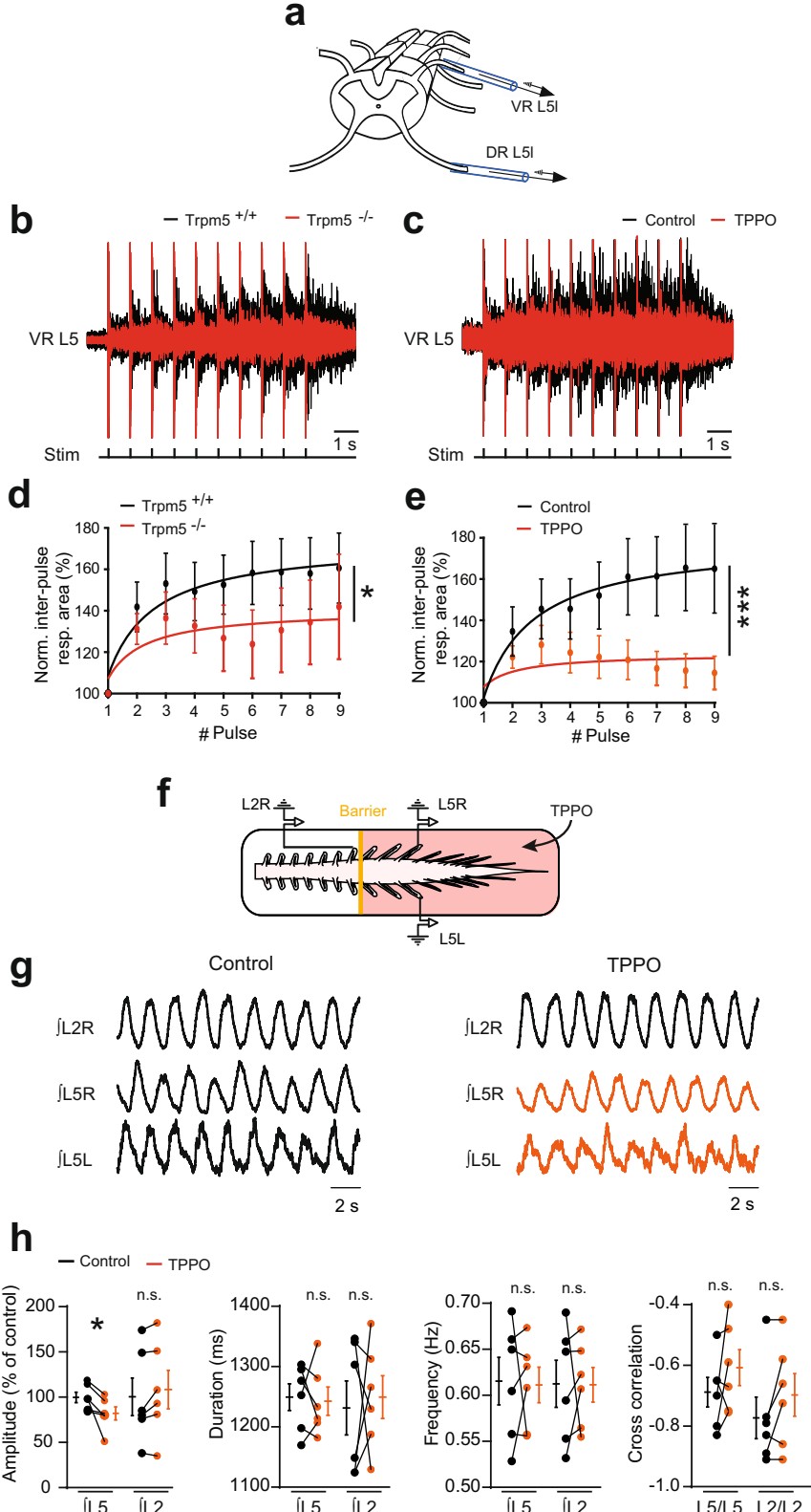

Digidata 1322A interface using the Clampex 10.3 software (Molecular Devices). All experiments were designed to gather data within a stable period (i.e., at least 2 min after establishing whole-cell access).

*Whole spinal cord preparation.* For the whole spinal cord preparation, motor outputs were recorded from lumbar ventral roots by means of glass suction electrodes connected to an AC-coupled amplifier. Electrode placed in contact with the dorsal roots was used to deliver repetitive (1 Hz) supramaximal stimuli (50–60 μA, 0.2 ms duration). The ventral root recordings were amplified (×2000), high-pass filtered at 70 Hz, low-pass filtered at 3 kHz, and sampled at 10 kHz. Custom-built amplifiers enabled simultaneous online rectification and integration (100 ms time constant) of raw signals. Locomotor-like activity was induced by a bath application of NMA (10 μM) and 5-HT (5 μM). In some experiments, a Vaseline barrier was built at the $L_2/L_3$ level to superfuse the locomotor network located in the rostral lumbar cord independently from the more caudally located motoneurons.

**Fig. 6 Trpm5 channels amplify motor outputs. a** Schematic representation of the ventral spinal cord side up with the stimulating (DR L5L) and recording (VR L5L) glass electrodes. **b, c** Ventral root (L5) responses to 1-Hz ipsilateral dorsal root stimuli, recorded from wild-type ($n = 11$ mice) and Trpm5$^{-/-}$ ($n = 7$ mice) spinal cords (**b**) or from wild-type spinal cords before and after bath-applying triphenylphosphine oxide (TPPO, 30 µM, $n = 8$ mice) (**c**). **d, e** Quantification of the response as a function of the pulse number. Values are relative to the area of the initial post-stimulation response measured during the first inter-pulse interval. **f** Schematic representation of the whole-mount spinal cord with the recording glass electrodes from the ipsilateral (L2R, L5R) and contralateral (L5L, L5R) sides. The yellow solid line represents the Vaseline barrier separating the rostral (L2) from the caudal (L5) segments. **g** Ventral root recordings of NMA/ 5-HT-induced rhythmic activity generated before and after adding triphenylphosphine oxide (TPPO, 30 µM, $n = 6$ mice) to caudal lumbar segments. **h** Quantification of locomotor burst parameters. n.s., no significance; *$P < 0.05$; ***$P < 0.001$ (fit comparison for **d, e**; two-tailed Wilcoxon paired test for **h**). Mean ± SEM. For detailed $P$ values, see Source data. Source data are provided as a Source data file. See also Supplementary Fig. 7.

## Assessment of motor behaviors

*Walking*. The CatWalkXT (Noldus Information Technology, Netherlands) was used to measure walking performance. Each animal walked freely through a corridor on a glass walkway illuminated with beams of light from below. A successful walking trial was defined as having the animal walk at a steady speed (no stopping, rearing, or grooming), and three to five successful trials were collected per animal. The footprints were recorded using a camera positioned below the walkway, and footprint classification was manually corrected to ensure accurate readings. The paw print parameters were then analyzed using the CatWalk software (see data analysis).

*Surface righting reflex*. In this test, animals were maintained in a supine position for ~1 s and then released. The performance corresponded to the time spent to recover a complete prone position (four paws in contact with the plane testing surface with a cutoff period fixed at 120 s). It was recorded during at least two consecutive trials spaced by a ~30 s interval.

*Rotarod test*. Mice were placed on a rotarod (Bioseb) rotating at a fixed-speed (5, 10, 15, and 20 rpm) or accelerating from 4 to 40 rpm over a span of 5 min. Mice were given 3 trials with a 30-s inter-trial interval.

*Swimming*. Mice were gently placed individually in the center of the tank (18900 cm³) filled with heated water (30–33 °C). Swimming distance and velocity were quantified during three consecutive 15, 30, 90, and 120 s periods for P5, P9, P12, and adult mice, respectively. Each trial was spaced by a 5–15 min interval. At the end of the trial, the mouse was immediately removed from the tank, dried off with a paper towel, and returned to its homecage. Swimming parameters tracking and analysis were performed by using an automated video-tracking Ethovision system (Noldus). All behavioral experiments were carried out with the experimenter blind to genotype.

## Immunohistochemistry

Spinal cords of 10–12-day-old mice were dissected out and fixed for 5–6 h in 4% paraformaldehyde, then rinsed in phosphate-buffered saline (PBS), and cryoprotected overnight in 20% sucrose at 4 °C. Spinal cords were frozen in OCT medium (Tissue Tek) and 30 µm cryosections were collected from the L4–L5 segments. After having been washed in PBS 3 × 5 min, the slides were incubated for 1 h in a blocking solution (bovine serum albumin 1%, normal donkey serum 3% in PBS) with 0.2% triton X-100 and for 48 or 12 h at 4 °C in a humidified chamber with the primary antibody anti-ChAT (choline acetyltransferase; Millipore AB144P from goat) or with the primary antibody anti-GFAP (glial fibrillary acidic protein, AgilentDako Z0334 from rabbit), respectively. Both antibodies were diluted in the blocking solution with 0.2% triton X-100 (1:100 and 1:1000 for anti-ChAT and anti-GFAP, respectively). Slides were then washed 3 × 5 min in PBS and incubated for 2 h with an Alexa Fluor® Plus 555-anti-goat IgG secondary antibody (Invitrogen, A32816 from donkey) diluted (1:400) in the blocking solution. After 3 washes of 5 min in PBS, they were mounted with a gelatinous aqueous medium. Images were acquired using a confocal microscope (LSM700, Zeiss) equipped with a ×40 oil objective and processed with the Zen 2.6 software (Zeiss).

## TRPM5 mRNA quantification

*Cell culture*. HEK293 cells were transiently cotransfected with a plasmid encoding mouse Trpm5 (VB2010603-1201mug, VectorBuilder) plus a plasmid encoding shRNA Trpm5 or control using Lipofectamine 3000 (ThermoFisher). Cells were harvested 48 h after transfection and homogenized with QIAshredder spin column (Qiagen).

*Spinal cord*. The lumbar part of the spinal cord was dissected in aCSF at 4 °C and conserved in RNAprotect Tissue Reagent (Qiagen) at −20 °C until RNA extraction.

*Quantitative RT-PCR*. Total RNA from cells and tissues was extracted using the RNeasy plus mini kit (Qiagen) and cDNA was generated with the SuperScript IV VILO Master Mix (ThermoFisher) from 10 ng initial RNA for HEK293 and 500 ng for spinal cord. Trpm5 expression was quantified with Taqman Gene Expression Assays (Trpm5 (Mm01129032_m1), GAPDH (Hs99999905_m1), ACTB

(Mm01205647_g1)) and TaqMan Gene Expression Master Mix on a QuantStudio 7 (ThermoFisher). The relative expression of Trpm5 was calculated using the $2 - \Delta\Delta Ct$ method with GAPDH (cells) or ACTB (spinal cord) as internal reference. Results are expressed relative to control shRNA.

## TRPM5 protein quantification

*Membrane protein isolation and western blots*. Tissues were collected from spinal cord lumbar enlargements and frozen after removing the dorsal and ventral roots. For the membrane fraction, corresponding to the plasma membrane-enriched fraction, samples were homogenized in ice-cold lysis buffer (320 mM sucrose, 5 mM Tris-HCL pH 7.5, 10 µM iodoacetamide) supplemented with protease inhibitors (CompleteMini, Roche diagnostic Basel, Switzerland). Unsolubilized material was pelleted by centrifugation at $7000 \times g$ for 5 min. The supernatant was subjected to an additional centrifugation step at $18,000 \times g$ for 70 min at 4 °C. Pellets were collected and homogenized in ice-cold lysis buffer (1% Igepal CA-630, PBS 1×, 0.1% SDS, 10 µM iodoacetamide), supplemented with protease inhibitors (CompleteMini, Roche diagnostic). Protein concentrations were determined using a detergent-compatible protein assay (Bio-Rad). Equal protein amounts (30 µg) from samples were size fractionated by 4–15% Mini-PROTEAN TGX stain-free gels (Bio-Rad), transferred to a nitrocellulose membrane and probed with a polyclonal Trpm5 antibody (1:300, ACC-045, Alomone from rabbit) at 4 °C overnight in Tris-buffered saline containing 5% fat-free milk powder. The blot was then incubated for 1 h at 22 °C with a polyclonal horseradish peroxidase-conjugated anti-rabbit IgG secondary antibody (1:40,000; AB228341, ThermoFisher).

## Drug list and solutions

Normal aCSF was used in most cases for in vitro electrophysiological recordings. Ca²⁺-free solution was made by removing Ca²⁺ chloride from the recording solution and replacing it with an equimolar concentration of magnesium chloride. Low-Na⁺ solution was made by substituting equimolar concentrations of Na⁺ chloride by choline chloride. All solutions were oxygenated with 95% O₂/5% CO₂. All salt compounds, TEA (#T2265); TPPO (#T84603), L.A. (#L1376), BAPTA (#A9801); Caffeine (#C0750), NMA (#M2137), and 5-HT (#S2805) were obtained from Sigma-Aldrich. TTX (#1078), Chelerythrin (#1330), U73122 (#1268), Xestospongin-C (#1280), 9-Phenanthrol (#4999), Dantrolene (#0507), and Thapsigargin (#1138) were obtained from Tocris Bioscience. Dantrolene, thapsigargin, Xestospongin-C, U73122, and 9-Phenantrol were dissolved in dimethylsulfoxide (DMSO) and added to the aCSF (final concentration of DMSO: 0.05–0.1%). L.A. was dissolved in ethanol and added to aCSF (final concentration of ethanol: 0.05–0.1%). Control experiments showed no effects of the vehicle. The other drugs were dissolved in water and added to the aCSF.

## Neurophysical models

Modeling was employed as an independent test for the conceptual mechanism formed in this paper. Computational experiments were performed in NEURON simulation environment on our modified multi-compartment spinal motor neuron model[1]. The modification comprised in supplementing the soma with channels conducting L-type Ca²⁺ current ($J_{CaL}$) and a non-selective cationic current that had properties characteristic of Trpm5 current ($J_{TRPM5}$). Equations and parameters for all currents are given below and in the Table 1. Unlike earlier models representing Trpm5 as a non-specific inward current, here we explicitly define its ionic components based on experimental estimates of the channel's relative permeability (see below).

The Trpm5 model was a modification of our earlier model[66]. The latter described the channel voltage and temperature gating by the equations derived from a two-state kinetic scheme (Eqs. 12 and 13 in ref. [67]) with the parameters fitted to experimental data for Trpm5 [Fig. 1d, e in ref. [35]]. The channel opening sensitivity to intracellular calcium concentration [Ca²⁺]ᵢ was described by the Hill equation with the parameters fitted to experimental data [Fig. 8b in ref. [68]]. The current equation is:

$$J_{TRPM5} = G_{TRPM5} \cdot m_{V,T} \cdot m_{Ca} \cdot (E - E_{TRPM5})$$

where $G_{TRPM5}$ is the maximum conductivity; $m_{V,T}$ and $m_{Ca}$ are the kinetic variables of, respectively, voltage/temperature and Ca²⁺ dependence; $E$ is the transmembrane potential; and $E_{TRPM5}$ is the reversal (equilibrium) potential. The

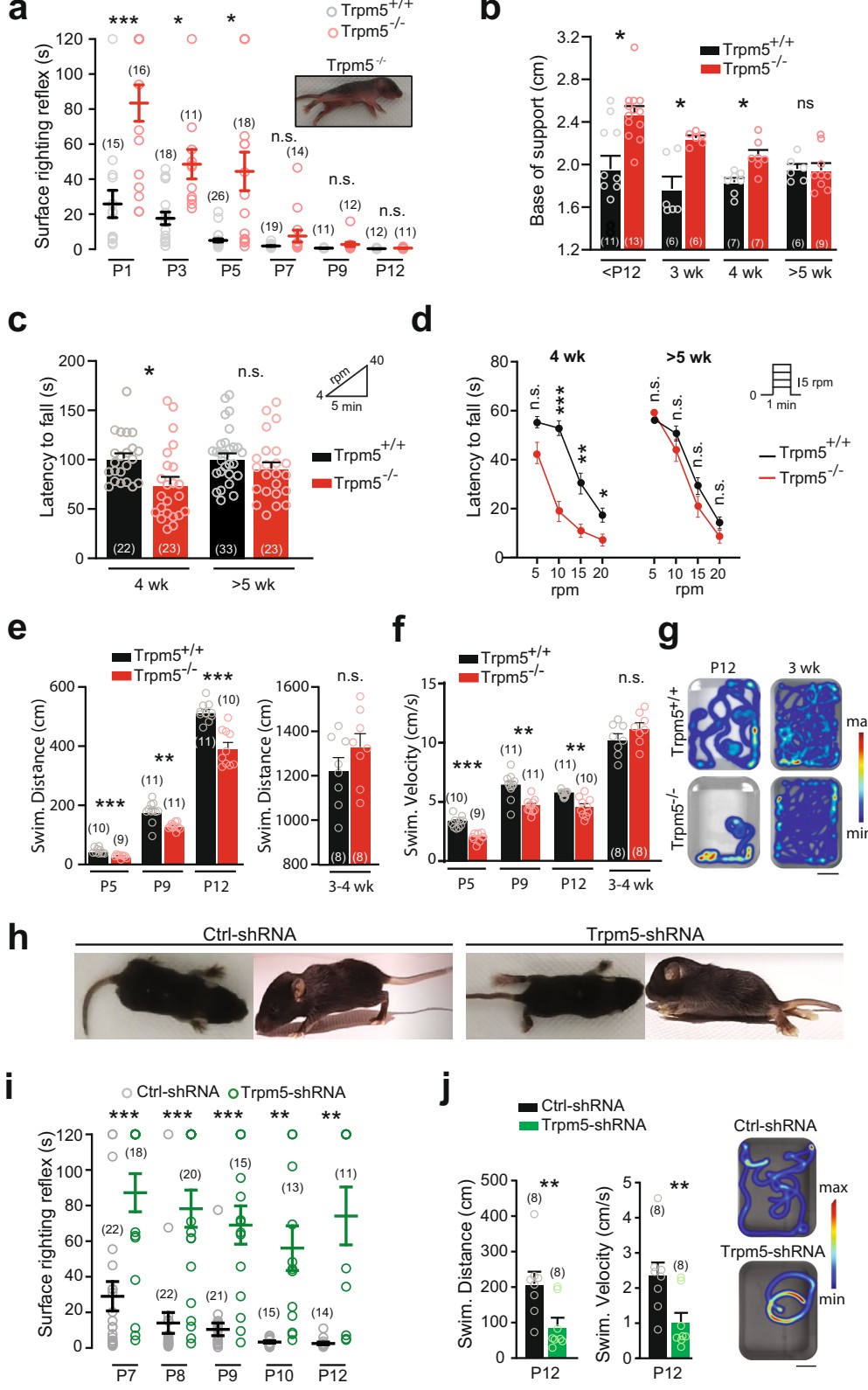

$m_{V,T}$ variable is described as

$$m_{V,T} = 1/(1 + \exp(-z \cdot F \cdot (E - E_{1/2})/RT)),$$

where $z$ is effective valence of the gating voltage-sensor; $E_{1/2} = (\Delta H - T \cdot \Delta S)/z \cdot F$ is the membrane potential of the channels half-activation; $F$ and $R$ are the Faraday and gas constants, respectively; $T$ is absolute temperature; $\Delta H$ and $\Delta S$ are, respectively, changes in enthalpy and entropy during the channel transition

between open and close states. The $m_{Ca}$ variable obeyed the differential equation

$$dm_{Ca}/dt = (m_{Ca,\infty} - m_{Ca})/\tau_{mCa},$$

where $\tau_{mCa}$ is the time constant of $m_{Ca}$ relaxation to its steady value $m_{Ca,\infty}$ described by the Hill equation $m_{Ca,\infty} = 1/(1 + (K_{1/2}/[Ca^{2+}]_i)^H)$; $K_{1/2}$ is the $[Ca^{2+}]_i$ value for the channels half-activation; $H$ is the Hill coefficient (cooperativity factor).

**Fig. 7 Trpm5 channels ensure motor control of hindlimbs. a** Surface righting response as a function of postnatal day in wild-type (black) and Trpm5$^{-/-}$ (red) mice. Values represent the time spent for rotating from a supine position to a prone position on their four paws. Picture illustrates a Trpm5$^{-/-}$ mouse that fails to right itself within 2 min. **b** Quantification of the base of support between hindlimb paws during walking as a function of age in wild-type (black) and Trpm5$^{-/-}$ (red) mice. **c** Latency to fall from a rod rotating at accelerated speed (4–40 rpm) in young adult mice (4 weeks and >5 weeks old), either wild-type (black) or Trpm5$^{-/-}$ (red). **d** Latency to fall from a rod rotating at constant speed in young adult mice (4 weeks and >5 weeks old), either wild-type (black) or Trpm5$^{-/-}$ (red). **e, f** Mean swimming traveled distance (**e**) and velocity (**f**) of neonates (P5–P12) and young adult (3–4 weeks old) wild-type (black) and Trpm5$^{-/-}$ (red) mice during three consecutive trials. **g** Heatmap representation of the swimming of neonates (P12) and young adult (3 weeks old) in wild-type (top) and Trpm5$^{-/-}$ mice (bottom). Scale bar, 10 cm. **h** Top and side views of 12-day-old wild-type mice transduced either with the scramble shRNA (left) or with the Trpm5-shRNA (right). **i** Surface righting response during postnatal development in wild-type mice transduced either with the scramble shRNA (black) or with a Trpm5-shRNA (green). **j** Swimming activity of 12-day-old wild-type mice transduced either with the scramble shRNA (black) or with the Trpm5-shRNA (green) Left: Swimming distance and velocity were averaged from three consecutive swimming trials. Right: Heatmaps illustrate swimming activity. Scale bar, 10 cm. The numbers in the brackets indicate the numbers of mice. n.s., no significance; *$P < 0.05$; **$P < 0.01$; ***$P < 0.001$ (two-way ANOVA with Sidak's multiple comparaisons test for **a–i**; two-tailed Mann–Whitney test for **j**). Mean ± SEM. For detailed $P$ values, see Source data. Source data are provided as a Source data file. See also Supplementary Fig. 8.

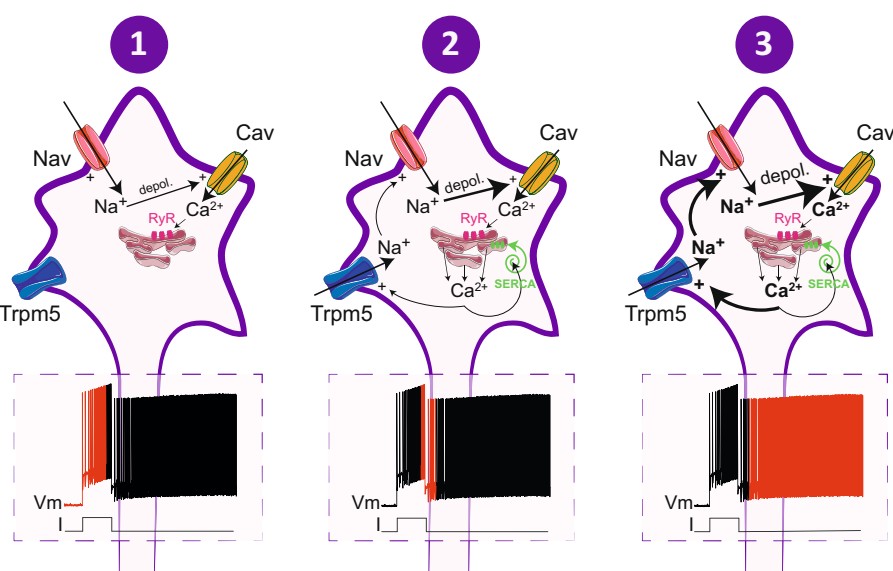

**Fig. 8 Overview of ionic cascades leading to bistability in spinal motoneurons.** Schematic relationships between currents underlying the different phases of the self-sustained firing mode. Trpm5 transient receptor potential cation channel subfamily M member 5, Nav voltage-gated sodium channels, Cav voltage-gated calcium channels, depol depolarization, RyR ryanodine receptors, Serca sarco/endoplasmic reticulum Ca$^{2+}$-ATPase.

The modifications of the above model comprised in decomposition of the total $J_{TRPM5}$ current into three monovalent ion components, Na$^+$, K$^+$, and Cs$^+$ ($J_{TRPM5} = J_{TRPM5,Na} + J_{TRPM5,K} + J_{TRPM5,Cs}$), for which the reported channel relative permeability is approximately equal and significantly exceeds that to Ca$^{2+}$ and Mg$^{2+}$, namely $P_{Na}:P_K:P_{Cs} = 1{:}1{:}1 \gg P_{Ca}$, ($P_{Ca}/P_{Na} = 0.05$)[36,46,47,68]. For simplicity, we assumed holding the conditions for the partial permeability ratio to equal the ratio of partial conductivities. This allowed representing $G_{TRPM5}$ as the sum of equal components:

$$G_{TRPM5} = G_{TRPM5,Na} + G_{TRPM5,K} + G_{TRPM5,Cs},$$

$$G_{TRPM5,Na} = G_{TRPM5,K} = G_{TRPM5,Cs} = G_{TRPM5}/3.$$

We kept the reversal potentials for Na$^+$ and K$^+$ currents the same as in the earlier motor neuron model[1], i.e. $E_{Na} = 45$ mV and $E_K = -85$ mV, and set $E_{TRPM5} = 0$ mV close to reported values. Consequently, expression of $E_{TRPM5}$ in terms of partial reversal potentials

$$E_{TRPM5} = (E_{Na} + E_K + E_{Cs})/3 = 0$$

required setting the unique free parameter $E_{Cs} = -(E_{Na} + E_K) = 40$ mV. Such $E_{Cs}$ is the Nernst potential at 35 °C for inner and outer concentrations of [Cs$^+$]$_I = 0.06377$ μM and [Cs$^+$]$_o = 0.2857$ μM. The latter corresponds to the reported median concentration of Cs$^+$ in the cerebrospinal fluid of healthy humans, 0.28 μM[69].

*Protocol of computational experiments.* A step-wise electrode current applied at the soma was a superposition of a background depolarization current (to shift the holding potential), a "stimulus" depolarization current (a step or pulses), and, in some cases, a hyperpolarization step (to test the true electrical bistability by toggling the membrane potential to downstate near-resting level). The membrane potentials were recorded from the soma at temperatures of 33 or 22 °C with intact

or "knocked-out" Trpm5 channels (Trpm5$^+$ and Trpm5$^-$, respectively). Trpm5$^-$ condition was simulated by setting zero maximum conductivity of the channels. The recorded sequences of action potentials were quantitatively characterized and graphically represented by momentary frequency and raster plots. A special set of computation experiments was performed on 50 uncoupled simulated motoneurons of the same geometry. Intra-population variability was simulated by randomization of $G_{TRPM5}$, which had Gaussian distribution with mean value 55 mS/cm$^2$ and standard deviation 11 mS/cm$^2$ (in some experiments, the conductivity was reduced by ~75%). All cells received the same synaptic excitation simulated by sinusoid (delay 1 s, 10 cycles at a frequency of 1 Hz) changes in the synaptic conductances inserted either in the soma (Fig. 5g, j) or in proximal dendritic branches (Fig. 5h, i, k, l), which had lengths equal to 78, 48, and 52 μm and were located at path distances of, respectively, 69, 54, and 31 μm from soma on different dendrites. The activity of the population was represented by a sequence of raster plots of individual cells spiking where each horizontal line represented a neuron and each small vertical bar represented a spike. From these data, the integrated population activity was derived and represented by the histogram of average number of spikes per second per neuron (bin width 20 ms).

*Model equations.* Ion currents in the modeled motoneuron were described by the Hodgkin–Huxley-type equations in terms of the $i$-th component current density per unit membrane area, $J_i$ (mA/cm$^2$):

$$J_i = G_i \cdot m^x \cdot h^y \cdot (E - E_i),$$

where $G_i$ is the maximum specific conductivity (S/cm$^2$); $m$ and $h$ are kinetic variables of, respectively, activation and inactivation with the respective orders $x$ and $y$; $E$ is the membrane potential (mV), $E_i$ is the current reversal potential (mV).

Kinetic variables ($p = m, h$) obeyed differential equations

$$dp/dt = \alpha_p(E) \cdot (1 - p) - \beta_p(E) \cdot p$$

or equivalent

$$dp/dt = (p_\infty(E) - p)/\tau_p(E),$$

where $p_\infty(E) = \alpha_p(E)/(\alpha_p(E) + \beta_p(E))$ and $\tau_p(E) = 1/(\alpha_p(E) + \beta_p(E))$ are, respectively, the steady-state value of $p$ and the time constant of activation or inactivation; $\alpha_p(E)$ и $\beta_p(E)$ are voltage-dependent (or calcium-dependent) forward and backward rate constants, respectively. For some partial currents, $p_\infty$ and $\tau_p$ were defined by approximating functions fitted to available experimental data (see equations below).

The dynamics of intracellular calcium concentration $[Ca^{2+}]_i$ was described in a simplified form:

$$d[Ca^{2+}]_i/dt = -J_{Ca}/(2 \cdot F \cdot \delta) - ([Ca^{2+}]_{i0} - [Ca^{2+}]_i)/\tau_r$$

where $J_{Ca}$ is the total calcium current density; $F$ is the Faraday constant, $\delta = 0.2\ \mu m$ is the thickness of the sub-membrane cytoplasm layer; $T_r = 2.82$ ms is the time constant of $[Ca^{2+}]_i$ relaxation to its basal level $[Ca^{2+}]_{i0} = 70$ nM due to united action of pumps, buffers, and diffusion from the sub-membrane layer to the bulk of cytosol.

The partial currents equations are given below and their parameters in Table 1. Fast inactivating tetrodotoxin-sensitive sodium current ($J_{NaTTX}$):

$J_{NaTTX} = G_{NaTTX} \cdot m^3 \cdot h \cdot (E - E_{Na})$;
$dm/dt = \alpha_m \cdot (1 - m) - \beta_m \cdot m$;
$dh/dt = \alpha_h \cdot (1 - h) - \beta_h \cdot h$;
$\alpha_m = 0.417 \cdot (E + 43)/(1 - \exp(-(E + 43)/5))$;
$\beta_m = 0.417 \cdot (E + 15)/(\exp((E + 15)/5) - 1)$;
$\tau_m = 0.5/(\alpha_m + \beta_m)$;
$\alpha_h = 0.3197/\exp((E + 65)/20)$;
$\beta_h = 4.6287/(\exp(-(E + 12.5)/10) + 1)$;
$\tau_h = 1/(\alpha_h + \beta_h)$.

Non-inactivating persistent tetrodotoxin-sensitive sodium current ($J_{NaP}$):

$J_{NaP} = G_{NaP} \cdot m^3 \cdot (E - E_{Na})$;
$dm/dt = (m_\infty - m)/\tau_m$;
$m_\infty = 1/(1 + \exp(-(E + 53)/6))$;
$\alpha_m = 0.011 \cdot (E + 21.4)/(1 - \exp(-(E + 21.4)/5))$;
$\beta_m = 0.00026 \cdot (E + 25.7)/(\exp((E + 25.7)/5) - 1)$;
$\tau_m = 1/(\alpha_m + \beta_m)$.

N-type calcium current ($J_{CaN}$):

$J_{CaN} = G_{CaN} \cdot c^2 \cdot d \cdot (E - E_{Ca})$;
$dc/dt = \alpha_c \cdot (1 - c) - \beta_c \cdot c$;
$\alpha_c = 0.2642 \cdot (19.88 - E)/(\exp((19.98 - E)/10) - 1)$;
$\beta_c = 0.064 \cdot \exp(-E/20.73)$;
$\tau_c = 1/(\alpha_c + \beta_c)$;
$dd/dt = \alpha_d \cdot (1 - d) - \beta_d \cdot d$;
$\alpha_d = 2.225 \cdot 10^{-4} \exp(-E/48.4)$;
$\beta_d = 1.39/(\exp((39 - E)/10) + 1)$;
$\tau_d = 1/(\alpha_d + \beta_d)$.

L-type calcium current ($J_{CaL}$):

$J_{CaL} = G_{CaL} \cdot e^2 \cdot (E - E_{Ca})$;
$de/dt = \alpha_e \cdot (1 - e) - \beta_e \cdot e$;
$\alpha_e = 15.69 \cdot (81.5 - E)/(\exp((81.5 - E)/10) - 1)$;
$\beta_e = 0.29 \cdot \exp(-E/10.86)$;
$\tau_e = 1/(\alpha_e + \beta_e)$.

Fast delayed rectification potassium current ($J_{fKDR}$):

$J_{fKDR} = G_{fKDR} \cdot n^4 \cdot (E - E_K)$;
$dn/dt = \alpha_n \cdot (1 - n) - \beta_n \cdot n$;
$\alpha_n = -0.097 \cdot (E + 23)/(\exp(-(E + 23)/6) - 1)$;
$\beta_n = 0.368 \cdot \exp(-(E + 48)/40)$;
$\tau_n = 1/(\alpha_n + \beta_n)$.

Rapidly activating slowly inactivating Kv1.2-type potassium current ($J_{Kv1.2}$):

$J_{Kv1.2} = G_{Kv1.2} \cdot k \cdot l \cdot (E - E_K)$;
$dk/dt = (k_\infty - k)/\tau_k$;
$dl/dt = (l_\infty - l)/\tau_l$;
$k_\infty = 1/(1 + \exp(-(E + 46)/6.9))$;
$\tau_k = 2.44 + 18.387/(\exp(-(E - 25.645)/21.633) + \exp((E + 4.42)/45.9))$;
$l_\infty = 1/(1 + \exp((E + 54)/7.1))$;
$\tau_l = 3.737/(0.00015 \cdot \exp(-(E + 13)/15) + 0.06/(1 + \exp(-(E + 68)/12)))$.

Small conductance calcium-dependent potassium current ($J_{SK}$):

$J_{SK} = G_{SK} \cdot q^2 \cdot (E - E_K)$;
$dq/dt = \alpha_q \cdot (1 - q) - \beta_q \cdot q$;
$\alpha_q = 0.00246/\exp(-(12 \cdot \log([Ca^{2+}]_i) + 28.48)/4.5)$;
$\beta_q = 0.006/\exp((12 \cdot \log([Ca^{2+}]_i) + 60.4)/35)$.

Leakage current ($J_{Leak}$):

$J_{Leak} = G_{Leak} \cdot (E - E_{Leak})$.

Excitatory synaptic current ($J_{syn}$)

$$J_{syn} = G_{syn} \cdot (E - E_{syn})$$

$$G_{syn} = G_{synmax} \cdot (1 + \sin(2 \cdot \pi \cdot t \cdot f + \pi/2))/2,$$

where $t$ is time (s), $f$ is frequency (1/s), and $G_{synmax}$ (S/cm$^2$) is maximum synaptic conductivity.

Voltage- and temperature-gated calcium-sensitive TRP current TRPM5 ($J_{TRPM5}$):

$$J_{TRPM5} = G_{TRPM5} \cdot m_{V,T} \cdot m_{Ca}(E - E_{TRPM5});$$
$$m_{V,T} = 1/(1 + \exp(-z \cdot F \cdot (E - E_{1/2})/R \cdot T));$$
$$E_{1/2} = (\Delta H - T \cdot \Delta S)/z \cdot F;$$
$$dm_{Ca}/dt = (m_{Ca,\infty} - m_{Ca})/\tau_{mCa};$$
$$m_{Ca,\infty} = 1/(1 + (K_{1/2}/[Ca^{2+}]_i)^H);$$

where $z = 0.577$ (dimensionless), $\Delta H = 1.2e5$ (J), $\Delta S = 389$ (J/K), $K_{1/2} = 8e{-}3$ (mM), $H = 3.2$ (dimensionless), and $\tau_{mCa} = 150$ (ms).

Noteworthy, for all currents except $J_{SK}$ and $J_{TRPM5}$ currents the time constants in the differential equations of activation and inactivation kinetic variables were divided by the temperature coefficient $q_{10} = 3^{((t°-33)/10)}$, where $t°$ is temperature in °C.

The equilibrium (reversal) potentials for partial sodium, potassium, leakage, and synaptic currents were fixed: $E_{Na} = 45$ mV; $E_K = -85$ mV; $E_{Leak} = -70$ mV, $E_{syn} = 0$ mV. For calcium currents, this potential varied due to changes in $[Ca^{2+}]_i$ as defined by the Nernst equation $E_{Ca} = (RT/2\ F) \cdot \ln([Ca^{2+}]_i/[Ca^{2+}]_o)$ with assumed constant extracellular concentration $[Ca^{2+}]_o = 2$ mM. For TRPM5 current, $E_{TRPM5} = 0$ mV was the weighted sum of partial equilibrium potentials of its Na$^+$, K$^+$, and Cs$^+$ components: $E_{TRPM5} = (E_{Na} + E_K + E_{Cs})/3$. The common weighting factor of (1/3) was determined by the assumed equality of the maximum partial conductivities in the total conductivity:

$$G_{TRPM5} = G_{TRPM5,Na} + G_{TRPM5,K} + G_{TRPM5,Cs;}$$
$$G_{TRPM5,Na} = G_{TRPM5,K} = G_{TRPM5,Cs} = G_{TRPM5}/3.$$

With the given $E_{Na}$ and $E_K$, the value $E_{TRPM5} = 0$ mV was provided by setting $E_{Cs} = 40$ mV. Correspondingly, specific components of the total $J_{TRPM5}$ current were $J_{TRPM5,Na} = (1/3) \cdot G_{TRPM5} \cdot (E - E_{Na})$, $J_{TRPM5,K} = (1/3) \cdot G_{TRPM5} \cdot (E - E_K)$, $J_{TRPM5,Cs} = (1/3) \cdot G_{TRPM5} \cdot (E - E_{Cs})$.

**Data quantification.** Electrophysiological data analyses were analyzed off-line with the Clampfit 10.7 software (Molecular Devices). For intracellular recordings, several basic criteria were set to ensure optimum quality of intracellular recordings. Only cells exhibiting a stable resting membrane potential, access resistance (no >20% variation), and an action potential amplitude >40 mV under normal aCSF were considered. Passive membrane properties of cells were measured by determining from the holding potential the largest voltage deflections induced by small current pulses that avoided activation of voltage-sensitive currents. We determined input resistance by the slope of linear fits to voltage responses evoked by small positive and negative current injections. Firing properties were measured from depolarizing current pulses of varying amplitudes. The rheobase was defined as the minimum step current intensity required to induce an action potential from the membrane potential held at $V_{rest}$. Peak spike amplitude was measured from the threshold potential, and spike duration was measured at half-amplitude. The instantaneous discharge frequency was determined as the inverse of interspike interval. All reported membrane potentials were corrected for liquid junction potentials (+14 mV). In TTX/TEA condition, the peak amplitude of the sADP was defined as the difference between the holding potential $\sim{-}65$ mV and the peak voltage deflection after the burst of spikes. The sADP area was measured between the end of the stimulus pulse and the onset of the hyperpolarizing pulse (delta = 7.5 s). For a direct comparison of sADP peak amplitudes before and during a drug application, current pulses were adjusted to reach the same level of calcium spiking activity during the pulse duration as that for control. If necessary, using bias currents, the pre-pulse membrane potential was maintained at the holding potential fixed in the control condition ($\sim{-}65$ mV). In normal aCSF, the summation (windup) of sADPs following repetitive depolarizing current pulses was measured as the difference in the peak amplitude of the first vs the last sADP. Traces with depolarizing current pulses of the same amplitude were chosen to compare the effect of drugs on both the amplitude and the windup of the sADP. Bistable properties were investigated with a 2 s depolarizing current pulses of varying amplitudes (1–3 nA). For 2 s before injecting the depolarizing pulse, incrementing holding currents (steps of 25 pA) were injected into the cell until reaching the spiking threshold. The cell was considered as bistable when (1) the pre-stimulus membrane potential stays relatively hyperpolarized below the spiking threshold (downstate), (2) the post-stimulus membrane potential stays relatively depolarized (upstate), at perithreshold for spike generation, and (3) the membrane potential switches to downstate after a brief hyperpolarizing pulse. Thus, the ability of a motoneuron to be bistable was estimated by quantifying the difference ($\Delta V$) between the most hyperpolarized holding potential ($V_h$ min) at which the motoneuron can generate a self-sustained spiking and the most depolarized holding potential ($V_h$ max) at which the motoneuron can still generate a self-sustained spiking (see Supplementary Fig. 1c, d). For extracellular recordings, alternating activity between right/left L5 recordings was taken to be indicative of fictive locomotion. To characterize locomotor burst parameters, raw extracellular recordings from ventral roots were rectified, integrated, and resampled at 50 Hz.

Peak amplitude of locomotor burst was measured and the cycle period was calculated by measuring the time between the first two peaks of the auto-correlogram. The coupling between right/left L5 was estimated by measuring the correlation coefficient of the cross-correlogram at zero phase lag. In response to dorsal root stimuli, motor outputs were rectified integrated and smoothed with a time constant of 0.1 s. Area of ventral root activities were measured with threshold-based event detection. Mean response was constructed from three consecutive rectified responses. We computed responses at each interpulse interval.

**Behavioral analyses**. For walking, the CatWalkXT software (Noldus Information Technology, Netherlands) was used to measure a broad number of spatial and temporal gait parameters in several categories. These include (i) dynamic parameters related to individual paw prints, such as duration of the step cycle with the respective duration of the swing and stance phases; (ii) parameters related to the position of paw prints with respect to each other, for example, the stride length (distance between two consecutive placement of the same paw) and the base of support (the width between the paw pairs); (iii) parameters related to time-based relationships between paw pairs, as well as step patterns. These parameters were calculated for each run and for each paw. In the rotarod test, the average latency of the subject to fall was recorded (in seconds). A video tracking system (Noldus Information Technology, Wageningen, The Netherlands) was used to measure swimming performance. We ensured that there was a smooth tracking curve and that the center point of the animal remained stable before analysis took place. Distance and velocity during the test periods were scored. All behavioral tests were carried out with the experimenter blind to genotype.

**Statistics**. No statistical method was used to predetermine sample size. Group measurements were expressed as mean ± SEM. When two groups (control vs transgenic mice) were compared, we used the unpaired Mann–Whitney test. Fisher test was used to compare the percentages of bistability. When two conditions (control vs drugs) were compared, we used Wilcoxon matched pairs test. We also used a one-way or two-way analysis of variance tests for multiple comparisons. For all statistical analyses, the data met the assumptions of the test and the variance between the statistically compared groups was similar. The level of significance was set at $P < 0.05$. Statistical analyses were performed using the Graphpad Prism 7 software. They are indicated in the figure legends.

**Reporting summary**. Further information on research design is available in the Nature Research Reporting Summary linked to this article.

## Data availability

All data supporting the findings of this study are provided within the paper and its Supplementary information. Source data are provided with this paper.

## Code availability

Full sets of software files used for obtaining modeling results are available from the authors on request.

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

## Acknowledgements

We are grateful to Dr Michael Caterina, Aziz Mocqrich, Dr Pierre Launay, and Dr Robert F. Margolskee for their generous gift of transgenic Trpv2$^{-/-}$, Trpv1-3$^{-/-}$, Trpm4$^{-/-}$, and Trpm5$^{-/-}$ mice, respectively. We are grateful to Geneviève Rougon and Nejada Dingu for their critical reading of the manuscript, Cécile Brocard for extensive geno-typing and technical advices in Western blotting, and Anne Duhoux for animal care. This research has received supported from Agence National de la Recherche Scientifique (CalpaSCI, ANR-16-CE16-0004 to F.B.; nEURo*AMU, ANR-17-EURE-0029 to B.D.), the French Institut pour la Recherche sur la Moelle épinière et l'Encéphale (to F.B.), and the PRESTIGE program (PRESTIGE-2017-1-0006, PCOFUND-GA-2013-609102 to R.B.)

## Author contributions

R.B. designed and supervised the whole project, performed and analyzed the in vitro and in vivo experiments, and analyzed some immunohistochemical experiments. B.D. designed, performed, and analyzed the in vitro and in vivo experiments. M.B. designed and performed some of the in vitro experiments. E.P. designed, performed, and analyzed the immunohistochemistry and the molecular biology experiments and realized the AAV injections. V.T. designed, performed, and analyzed quantitative real-time PCR experiments on HEK cells. S.M.K. designed the cell model and performed the analysis. F.B. designed, supervised, and funded the whole project and performed and analyzed some in vivo experiments. R.B. and F.B. wrote the manuscript.

## Competing interests

The authors declare no competing interests.
