## [Peer Review File · Nature Communications]

TRPM5 CHANNELS ENCODE BISTABILITY OF SPINAL MOTONEURONS AND ENSURE MOTOR CONTROL OF HINDLIMBS IN MICEREVIEWER COMMENTS

Reviewer #1 (Remarks to the Author):

Summary of the study:

The important study under review investigates the origin of prolonged depolarization of motoneuron membrane (slow afterdepolarization, sADP) following steady depolarizing inputs. The sADP contributes to bistable behavior that has been shown to significantly influence motoneuron firing behavior. This motoneuron behavior is likely to be highly important in normal motor outflow and the results presented here constitute a major step forward. The study used spinal cord slices obtained from neonatal mice. In this preparation, depolarizing current pulses injected into motoneuron soma were used to evoke sADP. The data show that sADP is caused by a thermosensitive Ca^{2+} -activated Na^{+} current. Using a combination of pharmacological and genetic manipulations, the study elegantly shows that sADP is mediated by TRPM5 channels, and not by other thermosensitive channels (TRPV1,2,3 and TRPM4). The data also reveals the molecular pathway by which Ca^{2+} entry through voltage-gated Ca^{2+} channels stimulates further CICR via Ryanodine receptors which then causes activation of TRPM5 channels. Several pharmacological modulators were used to show the contribution of each part of this pathway, and also to test other pathways.

To test whether this pathway affects the neural output at the motor pool scale, the study used a whole-tissue spinal cord preparation. Both pharmacological and genetic silencing of TRPM5 caused reduction in the amplitude of ventral root output in response to repeated DR stimulation, as well as during fictive locomotion. The reduction in motor output was also predicted by a computational model of a motor pool.

At the behavioral level, data showed that TRPM5^{-/-} mice show postural motor deficits during early postnatal stages. Silencing channel expression in WT mice using shRNA revealed even more severe deficits that seemed to persist into later developmental stages.

All of the following comments are minor

1. The computational model of the motor pool:

The model does not fully capture all the experimental features of sADP, and, as presented, does not provide new information.

In the experimental data (fig 1-4), sADP seems to decay towards resting potential within few seconds, while the model shows that the simulated motoneurons continue to fire for more than a minute (Fig 5 a-d) until a hyperpolarizing current is injected. The reviewer suggests simulating a TTX-TEA experiment (Na^{+} and K^{+} channel conductance diminished) in an individual MN model, and generate a figure with the same scale as experimental data. Time constants of TRPM5 gating ($m\text{Ca}$) and $[\text{Ca}^{2+}]_i$ (τ_r) can be changed to match sADP kinetics in experimental data.

The raster plots in fig 5 e-f (top panel) are small, so individual dots representing spikes are not clear. In the middle panel, the maximum number of spikes/sec/neuron does not seem to change over the course of the train, the cumulative increase in the output seem to be driven by recruitment of new motoneurons rather than higher firing rates resulting from wind-up of sADP.

It is unclear why some motoneurons get recruited later in the train if stimulation pulses have the same amplitude.

It is appropriate while validating the model to use similar protocols as those of experimental data, such as a holding current to keep the cells around -60 mV and a square pulse to trigger firing. However, after validating the model, simulated physiological inputs (synaptic conductances) should be used to assess physiological behavior of the motor pool. This should be easy since the model already has multi-compartment cell models.

2. The contribution of Gq-coupled receptors to the activation of TRPM5:

Inhibitors of PLC, PKC, and IP3 receptors did not seem to affect the activity of TRPM5 channels in this study. However, in a slice preparation with no neuromodulators added the activity of these receptors is expected to be minimal, which is not necessarily the case in the intact spinal cord. The manuscript should discuss this possibility. This is crucial because neuromodulators have been historically shown to significantly enhance bistable behavior.

3. TRPM5 current as a component of persistent inward currents:

In the discussion section (line 307 and line 373), the manuscript presents the TRPM5 pathway as an alternative to the currently-held view of multi-component persistent inward currents mediating bistability. This view is not entirely accurate since motoneurons from TRPM5^{-/-} mice still express

sADP, although of smaller amplitude (fig 2c), ~50% of sADP amplitude is preserved in presence of TRPM5 blockers in WT cells (fig 2b), and 50% of TRPM5 $-/-$ motoneurons are still bistable (fig 4c). In addition, it is well known that a population of TTX-sensitive Na^+ channels contribute to bistability. In addition, in presence of neuromodulators such as 5-HT and NE (under physiological conditions), the persistent TTX-sensitive current and the L-type Ca^{2+} currents are enhanced.

4. Graphical presentation of data:

In order to make the figures easier to read and compare, the reviewer suggests the following:

In fig 3e and 3h, show the sADP amplitude and area, but not shown for the rest of the drugs in fig3 or other drugs in fig1 and fig2. It is better to keep it consistent across figures.

Switching the position of fig3 and fig4 may make the flow of the manuscript easier to follow.

Reviewer #2 (Remarks to the Author):

GENERAL COMMENTS

This project addresses an important issue for motor control: what are the intrinsic mechanisms by which spinal motoneurons produce and control posture, as well as influence locomotion? The authors present a novel mechanism that supports an old idea, namely that postural control requires plateau potentials and self-sustained firing patterns in alpha motoneurons. Here, the new mechanism posits that plateau potentials and sustained firing patterns utilize a temperature and Ca^{2+} -activated inward current mediated by Trpm5 ion channels.

The authors show that the mechanism involves Ca^{2+} mobilization from internal stores via Ryanodine receptors (and not protein kinase C, or inositol 1,4,5-trisphosphate receptors), which evokes Trpm5 inward currents that are able to sustain afterdepolarizations that become plateaus or sustained firing. The mechanism is interesting and unprecedented to the best of my knowledge.

The cellular electrophysiology is well done for the most part. However, a major flaw (see below) is that the developmental age of the slice preparations is never made clear: it could range from post-natal day 1 to 3 weeks age. That is an enormous range, and the properties of the motoneurons may change during that time.

The authors include a mathematical model, which supports but does not necessarily add a lot to their case. That is, the authors add the biophysical properties of Trpm5 to their existing motoneuron and the model behaves like their experimental recordings. The model does not make any non-intuitive testable predictions or explain the mechanism on a deeper level, it just recapitulates the experimental data.

The behavioral studies have two major shortcomings. First, the demanding form of locomotion that cannot be well executed without Trpm5 (measured via the rotarod task) is perfectly normal in the Trpm5 knockout mice by 5 weeks of age! That is astonishing! It suggests that the Trpm5 ion channel is only important in neonatal and juvenile stages (and not adult). I think there is a major developmental story missing here, and that the Trpm5 role may extinguish after juvenile stages of development. Second, the authors conclude that Trpm5 is for postural control. But there is a major contradiction in the data: stepping locomotion – which depends on postural control – is normal in Trpm5 knockout mice (at all ages studied). However, swimming – which is not a body weight-bearing form of exercise, and thus should not be as demanding from a postural control standpoint – shows a severe phenotype in the Trpm5 knockout mice at all ages studied. So, consider that demanding locomotion (e.g., rotarod) shows a phenotype, but only in neonates and juveniles, non-weight-bearing exercise swimming, reflex righting, but garden variety locomotion, which requires postural control, shows no phenotype ever. The base-of-support assay supports a postural phenotype, but the data are not strong given the mixed results recapped above.

So, while I like some aspects of the paper, there is a lack of clarity in several areas:

- What is the behavioral phenotype? Is it highly demanding locomotion like in the rotarod test? Is

it reflexes (as in righting reflex)? Or, is it strictly posture (i.e., base of support)?

- When does the phenotype manifest? Do the righting reflex and base of support problems also resolve by 5 weeks of age like rotarod?
- Why is a non-postural behavior like swimming impacted?
- Why does Trpm5 gene knockdown produce a stronger phenotype in general than Trpm5 gene knockout (see below)

There are some other major concerns regarding measurements and protocols, which are detailed below.

MAJOR CONCERNS

Supplementary Figure 1 provides a necessary description of the ΔV metric, calculated from V_{th} and V_{pp} , used throughout this study. Currently the description of ΔV is difficult to understand, and Supplementary Figure 1b only marginally helps with its interpretation. Perhaps it would help if Supplementary Figure 1b were incorporated into Figure 1, and furthermore, for Supplementary Figure 1b to clarify that V_{th} corresponds to the voltage of the sADP that is just below the threshold to evoke self-sustaining spiking. Consider re-coloring the dotted line for V_{th} so that it is distinguishable from the voltage traces and extending the dotted line to the sADP so the reader can see it. V_{pp} is defined as "...the most hyperpolarized...holding potential at which self-sustained spiking can be triggered". If V_{pp} is just the holding potential, then consider standard terminology of "holding potential" or V_h .

It is unclear how many mice were tested for shRNA knock down of Trpm5, and to what extent Trpm5 expression was attenuated. Lines 151-155 and Fig. 2e-g report eGFP expression in "80% of lumbar motoneurons". How many mice were tested? The authors specify 86 out of 107 cholinergic neurons but that does not tell the reader how many mice were tested. The "n" in Biology ought to be animal (Vaux, Fidler, & Cumming. EMBO reports 13(4): 291, 2012). One expects more than n=1 mouse tested. I would say n=1 mouse would be unacceptable.

More regarding shRNA knockdown: there is no quantification of the attenuation. There are a number of mechanisms to quantify the knockdown: Western blot for Trpm5 protein, quantitative PCR or multiplex in situ hybridization to measure Trpm5 transcripts quantitatively in AAV-shRNA-transduced vs. non-transduced motoneurons. One should quantify and report how much the shRNA actually knocked down the transcripts or attenuated protein expression. GFP expression only reports that the viral payload was dropped off and at least some of the payload was expressed.

One cannot compare the amplitude of ventral nerve output across Trpm5+/+ and Trpm5-/- mice. That occurs at line 263 with reference to Supplementary Fig. 7c. The comparison is misleading because suction electrodes can generate vastly different signals depending on the quality of the seal on the ventral root. The measurements themselves have no biophysical meaning (unlike intracellular recording). So, the amplitude can vary between different preparations including the root and electrode diameter (how tight is the fit?) as well as whether the glass is fresh or the electrode is being reused after cleaning. I am not convinced the measurements in Supp. Fig. 7c are meaningful or reliable. In contrast I trust the before/after comparison in Supp. Fig. 7b because TPPO is being applied to the same preparation and thus one can be confident the seal is the same before and after the drug is applied.

There are major developmental irregularities in the data, which are not explained. Consider the rotarod tests described in lines 276-283. Fig. 7d shows data from 4-week-old mice, in which the Trpm5-/- mice severely underperform compared to wild type. In contrast, Supp. Fig. 8g shows data from 5-week-old mice in which Trpm5-/- and wild type mice perform equally well. That is extraordinary compensation in 1 week. The locomotion in Trpm5-/- mice, and Trpm5-Trpm4 double knockout mice, is intact and matching at 3 weeks of age, thus I wonder about the base of support and righting reflex at 5 weeks and later adult stages? It seems that the later in development the animals are studied, the less phenotype they exhibit. Yet the only test measured as far out as 5 weeks is rotarod, and the Trpm5-/- mice are fine. The authors should also measure base of support and righting reflex out to 5 weeks too and determine whether the righting reflex

and base of support observations normalize at 5 weeks like rotarod performance. Perhaps all the phenotypes will be equally well compensated in the *Trpm5*^{-/-} mice or the *Trpm5*-*Trpm4* double knockout mice by 5 weeks. If so, it could be that *Trpm5* is only important in early postnatal development and has no real relevance in the adult. That developmental story could be interesting.

Regarding locomotion vs. swimming, the main take-home message of the paper is that *Trpm5* is important for postural control. Postural muscles must function during locomotion but there is no locomotor phenotype up to 3 weeks of age. However, the animals have an obvious swimming phenotype from P5-P12, shortly before 3 weeks of age. Postural control is not relevant to swimming, where the bodyweight is mostly supported by the buoyancy in water. It is hard to reconcile why there is a swimming phenotype -- but not a locomotor phenotype -- in *Trpm5*^{-/-} animals if the hypothesis that *Trpm5* is important for posture is correct.

The righting reflex data from *Trpm5*^{-/-} mice (Fig. 7a) and *Trpm5*-shRNA mice (Fig. 7f) are wildly variable. The righting reflex varies between ~3 sec to 120 sec in the experimental groups. That massive variability should be addressed. The control groups vary a lot too. Perhaps the authors should justify why this is still a good test when the animals vary so much. I have a hard time seeing signal through the noise. While I recognize one can always compute a mean but with this much variability I wonder if the data violate the assumptions of the central limit theorem, which precludes parametric statistics. The Methods specify the use of parametric and non-parametric tests, but in Figs. 7a & f it is not clear what test was applied, however I assume it would be an ANOVA for multiple levels of the independent variable.

The developmental stage of the in vitro cellular electrophysiology is unclear. The methods do not specify the age of the animals used, just "neonatal and young adult" but that is quite vague. Especially given the major changes in the righting reflex that occurs from P1 to P12, that behavior changes dramatically. One assumes that the motoneurons are also changing a lot during that developmental time period. It is essential that the authors specify the postnatal day of the preparations for cellular studies. Furthermore, the reader needs to know whether the properties measured so elegantly in Figures 1-4 hold steady during the entire range throughout which the behaviors sometimes change. Given the major performance improvement in rotarod at 5-weeks, I would like to see the motoneuron properties from 5-week-old slices (adult slice technologies are quite good now). The authors must document whether bistability and *Trpm5* generate the same or similar properties in motoneurons in adult stages older than 5 weeks by which time the *Trpm5*^{-/-} knockout mice no longer have a phenotype in the rotarod test.

It is unclear why (as reported on lines 293-295 and Fig. 7f) *Trpm5*-shRNA mice have such a severe locomotor deficit that they cannot even be tested on the rotarod after the 3rd postnatal week. As recapped above, *Trpm5*^{-/-} mice perform equally to wild-type mice by 5-weeks of age. So, why is there such a severe phenotype in the shRNA *Trpm5* knockdown? In general, shRNA knocks down gene expression by ~50%, whereas a gene knockout is complete (100%). So why should the phenotype in the *Trpm5*-knockdown be much more severe than in the *Trpm5*-knockout? It doesn't make sense unless we consider that germline knockout transforms the brain and CNS in which case we have to question all the in vitro data in Figs. 1-4 because those motoneurons would have been transformed through germline knockout.

In general, germline knockout mice are crude tools because (in this case) the gene is removed from the whole organism (*Trpm5* is expressed in more than the nervous system). It would have been advantageous to limit the knockout to motoneurons using a *Chat*-cre mouse crossed with a mouse featuring a LoxP-Stop-LoxP (LSL) floxed-allele of *Trpm5* to limit the *Trpm5* knockout to cholinergic motoneurons.

In Figure 5 and the text the authors describe a multi-compartment NEURON model which they use to test whether a *Trpm5* mediated conductance can underly the *I*_{CaN} current and reproduce the observed bistability and self-sustained firing in motor neurons. The overall results presented in Figure 5 are confirmatory but do not add any novel insights that could be subsequently tested in their in vitro experiments.

Figure 8 is unnecessary as it simply restates the interpretation of the results found on page 16, without adding anything further. It would be more suitable to rework Figure 8 into a graphical abstract.

MINOR (but still important) CONCERNS

Line 95: What is "large" in this context? Large compared to what? Do the authors have some cell dimensions for alpha motoneurons vs. (for example) Ia interneurons in an adjacent lamina?

Line 97: I do not agree that TEA at 10 mM will block "most" of the K⁺ channels. That concentration of TEA is on the low side. Yes, it will block some forms of K⁺ channels, but it will be insufficient for many others to be fully blocked. Therefore, I recommend that the authors instead write "Namely, after attenuating voltage-gated Na⁺ and K⁺ channels..." That rephrasing will be more accurate and does not impact how the authors interpret their data at all.

Lines 98-105: I think the authors should state that the spikes in Figure 1 in the presence of TTX/TEA are Ca²⁺ spikes and perhaps even include an inset to show the properties of a Ca²⁺ spike in TTX/TEA vs. a Na⁺ spike in control aCSF using a faster sweep speed so the reader could see the difference.

Line 107: Fig. 1h appears to emphasize temperature dependence of the plateau but not so much on brief supra maximal depolarization.

Line 112: Supplementary Fig. 1c is not much help because the motoneuron was tonically active to begin with. One cannot evoke a plateau or self-sustained firing from a tonic state.

Line 166: "Ins(1,4,5)P₃" should be formally defined. After writing it out they should define "IP₃" as the abbreviation throughout (e.g., Line 167). However, the authors also use "InsP₃" (Line 169). Please use only one abbreviation.

Lines 203-204: Please add the raw data points to Figs. 4c and 4d. All the other panes show the raw data points in addition to mean and standard deviation.

Line 207: Add raw data points for Supplementary Fig. 6c.

Line 227: What does "off" mean? Is G_{TRPM5} set to zero?

Reviewer #3 (Remarks to the Author):

The paper submitted by Bos et al builds on the group's prior paper (Bouhadjane et al 2013) which showed that bistability of motor neurons in neonatal rat is calcium-activated, sodium-mediated, and temperature sensitive, and suggested that the currents underlying bistability were mediated by TRPV2 channels. Here, they demonstrate that Trpm5 channels are the main carriers of the I_{CaN} which is important for mediating motor neuron bistability using pharmacology, knockout mice, and shRNA-mediated knockdown. Further, they go on to describe the mechanism of activation and inactivation of the Trpm5 by pharmacologically targeting potential regulators of intracellular calcium to determine which affect the slow afterdepolarization. Finally, genetic/shRNA-based silencing of Trpm5 channels is shown to lead to deficits in hindlimbs.

The findings will be of broad interest to the field. The L-type calcium channel has long been implicated in motor neuron bistability. Where this work does not directly contradict that, it suggests that the L-type calcium channel plays more of a supporting role, rather than directly underlying the slow depolarizing current. The study is comprehensive in terms of investigating mechanism of the sADP in detail. The experiments are well-considered and well-executed. Additionally, the manuscript is well-written and logically laid out. Overall, the figures are clear. It is an exciting study but there are some issues to be addressed:

1. The age of the animals and preparations should be made clear in every section. This is particularly the case for the motor neuron recordings. The Methods states "neonatal and young adult" (line 420). That is a large range and young adult is vague. Prior work of the group has shown differences in motor neuron properties within the first week of age (in rat). Additionally, the expression levels of L-type calcium channels are low in neonates.

2. The ΔV is used as a measure of "bistable ability" (line 108). The way in which this is calculated is not clear. For example, V_{th} is defined differently in the text and the legend. Is V_{th} the "most depolarized holding potential at which self-sustained firing can be triggered" (line 110) or the "spiking threshold" (i.e. line 1047, Supplementary line 10)? If the first, what does "th" stand for? If the second, is it the voltage threshold for action potential generation or the voltage of the cell at the rheobase step? The figure referred to when it is first defined (Supp Fig 1) is too condensed to be able to determine this. It is clear from the text (in several places) that V_{pp} is the most hyperpolarized voltage from which bistable firing can be triggered. However, the "pp" in V_{pp} is not obvious nor defined. Additional details as to how it is measured should be described (either in Results or Methods). Is it determined by 25pA steps? I would expect the V_{pp} to change based on current step amplitude and duration as well but at least the amplitude of the current step seems to vary, judging by the figures. What parameters are constant for comparison between cells?

3. In Fig 2e and f, the AAV-GFP expression looks to be exclusively in the motor neurons. The promoters (CMV or U6) are not targeting motor neurons. How widespread is the GFP (or shRNA) expression? If it is restricted to motor neurons, how is that the case? In lines 153-154, how many cords is the quantification of labeled motor neurons from and what is the variance cord to cord? For the recordings related to Figure 2, this information is not critical (because GFP+ motor neurons are being targeted) but this is important for whole cord experiments and behavioral outcomes in Figs 6-7. For example, how are effects seen in the top half of Fig 6 related to motor neurons exclusively?

4. The effects seen with the shRNA are dramatic so it is clearly having an effect but there is no validation of Trpm5 knockdown shown. A quantitative measure of the degree to which the channel is knocked down would be ideal. Also, due to the strong effects, particularly in the 3 week old mice in experiments related to Fig 7, are motor neurons still present/alive?

5. It is not clear why (or how) the data in Fig 7b, 7c, 7e, and 7h (and Supp Fig 8a, 8e, and 8f) are being normalized. The controls are not the same mice and individual data points are shown. Displaying the raw values for width, index, distance/set time, velocity, etc. on the y-axes would be more informative.

Minor

1. Please clarify if recordings were from lateral motor neurons or lateral and medial motor neurons.

2. It might be useful to see at least one set of Ca²⁺ and Na⁺ spikes on an expanded time scale. It was impossible to tell what V_{th} on Supp Fig 1 was referring to and one cannot see the differences in Fig 1f and g.

3. In addition to the reported n for the number of cells, please state the number of mice those cells came from, particularly for the experiments involving low numbers of recorded cells.

4. I suggest considering switching Figures 3 and 4 (and associated text). I think it would make for a more logical progression as Figures 2 and 4 are closely related.

5. Supplemental fig 3b: Means for instantaneous firing frequency do not appear to match the data. (It looks like a figure editing error since means are identical to panel a.)

6. Lines 281-3: This should be framed as a hypothesis as there are other possibilities (i.e. the channels are not important/relevant in adults).

7. Lines 411: I suggest rephrasing to be more similar to the last line of the Abstract, rather than "postural muscle activity".

8. The manuscript was very well written overall. There were just a few word choice errors and typos.

- Line 48: remove "downstream"
- line 49: spell out SERCA for first use
- line 50: remove "of"
- line 60: "in" should be "of"
- line 61: remove binary?
- line 75: "be" should be "being"
- line 106: "nearby" should be "near"
- line 144: "on" should be "to"
- line 145: "attested" to "confirmed"?
- line 206: should be "and were similar"
- line 274: "to walk" should be "from walking"
- line 286: "showed" should be "shown" (but showed is correct later in that same paragraph)
- line 332: "highest" should be "higher"
- line 341: should be "Furthermore, 2-APB directly inhibits.....but does not decrease...."
- line 479: intensity range?
- line 760: the "s" in sADP is slow, not small, right?
- line 776: "whether" should be "when"
- line 1034 and Supplemental lines 5-6: supra- and subliminal should be suprathreshold and subthreshold
- line 1067 and 1095: "irrelevant" should be "scramble" (if that is what it was) or "control"?
- Fig 1c x-axis: "no. spikes" or "# spikes"
- Fig 3h (right): y-axis should be area
- Fig 5e (bottom 2): y-axis should be mean
- Supp Fig 5c y-axis: ohms symbol was lost
- Supp Fig 6b: labels should be changed for clarity, perhaps "remained bistable in drug" and "no longer bistable in drug"?

REVIEWER COMMENTS

Reviewer #1 (Remarks to the Author):

Summary of the study:

The important study under review investigates the origin of prolonged depolarization of motoneuron membrane (slow afterdepolarization, sADP) following steady depolarizing inputs. The sADP contributes to bistable behavior that has been shown to significantly influence motoneuron firing behavior. This motoneuron behavior is likely to be highly important in normal motor outflow and the results presented here constitute a major step forward. The study used spinal cord slices obtained from neonatal mice. In this preparation, depolarizing current pulses injected into motoneuron soma were used to evoke sADP. The data show that sADP is caused by a thermosensitive Ca²⁺-activated Na⁺ current. Using a combination of pharmacological and genetic manipulations, the study elegantly shows that sADP is mediated by TRPM5 channels, and not by other thermosensitive channels (TRPV1,2,3 and TRPM4). The data also reveals the molecular pathway by which Ca²⁺ entry through voltage-gated Ca²⁺ channels stimulates further CICR via Ryanodine receptors which then causes activation of TRPM5 channels. Several pharmacological modulators were used to show the contribution of each part of this pathway, and also to test other pathways. To test whether this pathway affects the neural output at the motor pool scale, the study used a whole-tissue spinal cord preparation. Both pharmacological and genetic silencing of TRPM5 caused reduction in the amplitude of ventral root output in response to repeated DR stimulation, as well as during fictive locomotion. The reduction in motor output was also predicted by a computational model of a motor pool. At the behavioral level, data showed that TRPM5^{-/-} mice show postural motor deficits during early postnatal stages. Silencing channel expression in WT mice using shRNA revealed even more severe deficits that seemed to persist into later developmental stages.

All of the following comments are minor

1. The computational model of the motor pool:

The model does not fully capture all the experimental features of sADP, and, as presented, does not provide new information. In the experimental data (fig 1-4), sADP seems to decay towards resting potential within few seconds, while the model shows that the simulated motoneurons continue to fire for more than a minute (Fig 5 a-d) until a hyperpolarizing current is injected. The reviewer suggests simulating a TTX-TEA experiment (Na⁺ and K⁺ channel conductance diminished) in an individual MN model, and generate a figure with the same scale as experimental data. Time constants of TRPM5 gating (mCa) and [Ca²⁺]_i (τ) can be changed to match sADP kinetics in experimental data.

We thank Reviewer #1 for these helpful comments and suggestions. We modified the model according to the reviewer's suggestions. In simulations corresponding to TTX/TEA experiments, the conductances of both the transient (G_{NaTTX}) and persistent (G_{NaP}) Na⁺ currents and that of delayed rectifier K⁺ current (G_{KDR}) were diminished. In all computations, the time constant of calcium-dependent gating of *Trpm5* τ_{mCa} was changed (from 60 to 150 ms). The so modified model better captured the features of sADP in terms of both magnitude and duration observed in biological experiments under TTX/TEA conditions (cf. new fig.5 a-c compared to experimental figs.1b, 1g, and 2c). Continued self-sustained post-stimulus firing was generated by the modeled motoneuron when Na⁺ and K⁺ channels were intact (new fig.

5d-f), like in the prototypes under aCSF conditions (for instance cf. “control” cases in experimental figs. 3c, 3b, 4d, 4f).

To provide new information so far non-available in biological experiments, rather than to applied current pulses, we present the modeled motoneuron activities generated in response to synaptic excitation (new figs. 5g-l) of either soma or dendrites. Oscillatory synaptic excitation induced either a burst of spikes or triggered a self-sustained firing depending on *Trpm5* value. As a result, the integrated activity from an heterogeneous population of motoneurons built up approaching a steady level (Fig 5g,h middle and lower panels). The similar behaviour observed in response to synaptic excitation of the soma or distal dendritic branches evidenced robustness of the build-up. Interestingly, the known high expression of L-type Ca^{2+} channels on the soma and proximal dendrites in motoneurons (Zhang et al., 2006) may mediate Ca^{2+} entry in response to summed excitatory input to trigger the *Trpm5*-mediated plateau potential, as observed in the model. These new data are now illustrated in the figure 5, added in the result section (page 10 and 11) and discussed regarding the location of L-type Ca^{2+} channels (page 16, lines 14-20). The method section concerning the model has been updated.

The raster plots in fig 5 e-f (top panel) are small, so individual dots representing spikes are not clear.

To mark individual spikes in the raster plots of new fig 5 g-h (top panel), we now use thin short vertical bars instead of earlier used open circles. For a better comprehension of the signal, we also plotted inserts illustrating three distinct cases of firing (Fig. 5j-l).

In the middle panel, the maximum number of spikes/sec/neuron does not seem to change over the course of the train, the cumulative increase in the output seem to be driven by recruitment of new motoneurons rather than higher firing rates resulting from wind-up of sADP. It is unclear why some motoneurons get recruited later in the train if stimulation pulses have the same amplitude.

In the revised figs. 5g-l, we present responses of simulated motoneurons to cyclic synaptic excitation (somatic or dendritic) instead of those to periodic pulses of applied current. The quantitative characteristics of the responses are now computed and compared for each cycle that is more relevant because there are no distinct pulse and inter-pulse intervals anymore. The cycle-by-cycle wind-up of the population activity is due to both (a) increase in number of motoneurons switching to continuous self-maintained firing with synaptically modulated rate and (b) increase in the number of spikes with each consecutive 1-s cycle that means increase in the firing rate. Examples of (a) are given in figs. 5j,k with blue and green asterisks, whereas other panels illustrate (b) with red asterisk. A motoneuron transition to self-maintained firing in earlier or later cycle depends on randomized value of *Trpm5* conductance: the greater *Trpm5* the earlier the transition.

It is appropriate while validating the model to use similar protocols as those of experimental data, such as a holding current to keep the cells around -60 mV and a square pulse to trigger firing. However, after validating the model, simulated physiological inputs (synaptic conductances) should be used to assess physiological behavior of the motor pool. This should be easy since the model already has multi-compartment cell models.

To meet this request of the Reviewer #1 we performed computational experiments, in which sinusoidal changes in synaptic conductances were used as physiologically relevant inputs instead of periodic pulses of applied current. Such synaptic excitation was addressed either to the soma (new figs. 5g, 5j) or to several dendritic branches (new figs. 5h-i, 5k-l). In the illustrated case, these were 78- μm -, 48- μm -, and 52- μm -long branches located at path distances of, respectively, 69 μm , 54 μm , and 31 μm from soma on different dendrites.

2. The contribution of Gq-coupled receptors to the activation of TRPM5: Inhibitors of PLC, PKC, and IP3 receptors did not seem to affect the activity of TRPM5 channels in this study. However, in a slice preparation with no neuromodulators added, the activity of these receptors is expected to be minimal, which is not necessarily the case in the intact spinal cord. The manuscript should discuss this possibility. This is crucial because neuromodulators have been historically shown to significantly enhance bistable behavior.

We are fully agree with the reviewer and we have already discussed this point probably not clearly enough. In page 15 lines 17-20 we rephrased our sentence as follow “We showed in motoneurons that bradykinin recruits a Na^+ -mediated I_{CaN} via a similar signaling pathway⁵⁴. A possibility is that neuromodulators activating a G protein-coupled receptors recruit *Trpm5* channels.” As requested by the reviewer we also specified in the discussion that in our recording conditions the activation of the G protein-coupled receptors is expected to be minimal that could account for the insensitivity of the sADP to inhibitors of the PLC signaling cascade. (Page 15 lines 22-23).

3. TRPM5 current as a component of persistent inward currents: In the discussion section (line 307 and line 373), the manuscript presents the TRPM5 pathway as an alternative to the currently-held view of multi-component persistent inward currents mediating bistability. This view is not entirely accurate since motoneurons from TRPM5^{-/-} mice still express sADP, although of smaller amplitude (fig 2c), ~50% of sADP amplitude is preserved in presence of TRPM5 blockers in WT cells (fig 2b), and 50% of TRPM5^{-/-} motoneurons are still bistable (fig 4c). In addition, it is well known that a population of TTX-sensitive Na^+ channels contribute to bistability. In addition, in presence of neuromodulators such as 5-HT and NE (under physiological conditions), the persistent TTX-sensitive current and the L-type Ca^{2+} currents are enhanced.

Our data suggest a new point of view deciphering the ionic mechanisms involved in motoneurons bistability which remains consistent with previous works demonstrating the critical role of persistent inward currents (PICs) in bistability. The plateau potential is the sADP maintained by repetitive spiking. The persistent sodium current (I_{NaP}) does not play a significant role in generating the sADP because our recordings are performed under TTX (Fig. 1-2-4), but numerous studies demonstrated its critical role for generating the repetitive spiking in motoneurons (Kuo et al., 2006, J. Neurophy; Zhong et al., 2007; J. Neurosci; Pambo-Pambo et al., 2009, J. Neurophy). Therefore, I_{NaP} appears to play a critical role in bistability by its role in maintaining a spiking activity required for self-enforcing the plateau potential, but it does not play a direct role in mediating the TTX-insensitive plateau potential (sADP). Likewise for the persistent calcium current (I_{CaP}). We previously showed that the low-threshold activation of I_{CaP} plays a significant role in triggering the Ca^{2+} -induced Ca^{2+} -release mechanism required for recruiting secondary I_{CaN} which mediates a large part of the sADP (Bouhadfane et al., 2013). However, we also showed that I_{CaP} is dispensable for generating a normal sADP in motoneurons and can be supplanted by the recruitment of high-threshold voltage Ca^{2+} channels (Figure 5B in Bouhadfane et al., 2013). This demonstrates that this is not I_{CaP} per se which is important for generating sADP and plateau potential but the Ca^{2+} entry that it generates to recruit I_{CaN} . This important point is now added in the discussion (Page 16, lines 14-20). In line with this, the chelation of intracellular Ca^{2+} with BAPTA strongly affects the sADP without affecting Ca^{2+} spikes (Fig1f).

We agree with the reviewer that the blockade of TRPM5 channels did not fully abolish the sADP suggesting, as discussed in the manuscript (Page 15 lines 7-14), the contribution of

other channels. Regarding the strong effect of BAPTA we can posit the contribution of a second unidentified TRP channels, but our preliminary data show that the inactivation of Kv1.2 channels in motoneurons (Bos et al., 2018) significantly contributes to the remaining sADP. This specific point will be the subject of another publication.

We thus summarized in a graphical abstract the synergic ionic mechanisms involved in motoneurons bistability (Fig.8).

4. Graphical presentation of data:

In order to make the figures easier to read and compare, the reviewer suggests the following: In fig 3e and 3h, show the sADP amplitude and area, but not shown for the rest of the drugs in fig3 or other drugs in fig1 and fig2. It is better to keep it consistent across figures.

The figures are very loaded. There is a concern that adding the quantification of the sADP area in each panel will overload figures without providing more information. We plotted the sADP area specifically for Caffeine and thapsigargin (in Fig 4e and 4h) because these pharmacological compounds mainly modify the decay time constant of the sADP contrary to the other drugs which mainly affect the sADP amplitude.

Switching the position of fig3 and fig4 may make the flow of the manuscript easier to follow.

We thank the reviewer for this helpful comment which increases text comprehension. We thus switched the position of Fig 3 and Fig 4.

Reviewer #2 (Remarks to the Author):

GENERAL COMMENTS

This project addresses an important issue for motor control: what are the intrinsic mechanisms by which spinal motoneurons produce and control posture, as well as influence locomotion? The authors present a novel mechanism that supports an old idea, namely that postural control requires plateau potentials and self-sustained firing patterns in alpha motoneurons. Here, the new mechanism posits that plateau potentials and sustained firing patterns utilize a temperature and Ca²⁺-activated inward current mediated by Trpm5 ion channels.

The authors show that the mechanism involves Ca²⁺ mobilization from internal stores via Ryanodine receptors (and not protein kinase C, or inositol 1,4,5-trisphosphate receptors), which evokes Trpm5 inward currents that are able to sustain afterdepolarizations that become plateaus or sustained firing. The mechanism is interesting and unprecedented to the best of my knowledge.

The cellular electrophysiology is well done for the most part. However, a major flaw (see below) is that the developmental age of the slice preparations is never made clear: it could range from post-natal day 1 to 3 weeks age. That is an enormous range, and the properties of the motoneurons may change during that time.

The authors include a mathematical model, which supports but does not necessarily add a lot to their case. That is, the authors add the biophysical properties of Trpm5 to their existing motoneuron and the model behaves like their experimental recordings. The model does not

make any non-intuitive testable predictions or explain the mechanism on a deeper level, it just recapitulates the experimental data.

The behavioral studies have two major shortcomings. First, the demanding form of locomotion that cannot be well executed without Trpm5 (measured via the rotarod task) is perfectly normal in the Trpm5 knockout mice by 5 weeks of age! That is astonishing! It suggests that the Trpm5 ion channel is only important in neonatal and juvenile stages (and not adult). I think there is a major developmental story missing here, and that the Trpm5 role may extinguish after juvenile stages of development.

Second, the authors conclude that Trpm5 is for postural control. But there is a major contradiction in the data: stepping locomotion – which depends on postural control – is normal in Trpm5 knockout mice (at all ages studied).

However, swimming – which is not a body weight-bearing form of exercise, and thus should not be as demanding from a postural control standpoint – shows a severe phenotype in the Trpm5 knockout mice at all ages studied.

So, consider that demanding locomotion (e.g., rotarod) shows a phenotype, but only in neonates and juveniles, non-weight-bearing exercise swimming, reflex righting, but garden variety locomotion, which requires postural control, shows no phenotype ever. The base-of-support assay supports a postural phenotype, but the data are not strong given the mixed results recapped above.

So, while I like some aspects of the paper, there is a lack of clarity in several areas:

- What is the behavioral phenotype? Is it highly demanding locomotion like in the rotarod test? Is it reflexes (as in righting reflex)? Or, is it strictly posture (i.e., base of support)?
- When does the phenotype manifest? Do the righting reflex and base of support problems also resolve by 5 weeks of age like rotarod?
- Why is a non-postural behavior like swimming impacted?
- Why does Trpm5 gene knockdown produce a stronger phenotype in general than Trpm5 gene knockout (see below)

There are some other major concerns regarding measurements and protocols, which are detailed below.

MAJOR CONCERNS

Supplementary Figure 1 provides a necessary description of the ΔV metric, calculated from V_{th} and V_{pp} , used throughout this study. Currently the description of ΔV is difficult to understand, and Supplementary Figure 1b only marginally helps with its interpretation. Perhaps it would help if Supplementary Figure 1b were incorporated into Figure 1, and furthermore, for Supplementary Figure 1b to clarify that V_{th} corresponds to the voltage of the sADP that is just below the threshold to evoke self-sustaining spiking. Consider re-coloring the dotted line for V_{th} so that it is distinguishable from the voltage traces and

extending the dotted line to the sADP so the reader can see it. V_{pp} is defined as "...the most hyperpolarized...holding potential at which self-sustained spiking can be triggered". If V_{pp} is just the holding potential, then consider standard terminology of "holding potential" or V_h .

As requested by the reviewer we have significantly modified Suppl Fig. 1 and corrected/rephrased methods section for clarifying the measurements (page 31, lines 4-8). We now defined the most hyperpolarized and the most depolarized holding potential at which the motoneuron generate a self-sustained spiking as " V_h min" (rather than V_{pp} in the previous version) and " V_h max" (rather than V_{th} in the previous version), respectively. We also redefined ΔV in results section (page 5, lines 17-20).

It is unclear how many mice were tested for shRNA knock down of *Trpm5*, and to what extent *Trpm5* expression was attenuated. Lines 151-155 and Fig. 2e-g report eGFP expression in "80% of lumbar motoneurons". How many mice were tested? The authors specify 86 out of 107 cholinergic neurons but that does not tell the reader how many mice were tested. The "n" in Biology ought to be animal (Vaux, Fidler, & Cumming. EMBO reports 13(4): 291, 2012). One expects more than n=1 mouse tested. I would say n=1 mouse would be unacceptable.

To address more carefully the widespread expression in motoneuron pools, we performed additional quantifications in a total of 4 mice. The range of the transduction is mainly from T2-T3 to L5-S1. We did not observe any GFP+ expression in the supraspinal structures (starting from the brainstem to cortex) (Fig. S4) or cervical regions (not shown). We observed that 76 ± 3.6 % of lumbar motoneurons (227 out of 299 large cholinergic neurons analysed in the L4-L5 ventral horn, n = 4 mice), were fluorescent (Fig. 2g). However, a more careful analysis allowed us to see in the ventral horn some small non cholinergic cells transduced by the AAV and displaying complex processes extending from their cell bodies (see double arrow in Fig. 2h). Supplementary immunohistochemical experiments showed that they are GFAP positive suggesting that they are astrocytes. 19 ± 3.3 % of GFAP positive cells were transduced (122 out of 640 GFAP+ astrocytes were GFP+ from 4 mice). Our additional intracellular recordings confirmed the glial origin of these small cells (< 10 μ m) insofar as they were not capable of firing action potentials, displayed hyperpolarized resting membrane potential and low input resistance. Note that these electrophysiological properties of astrocytes transduced with a *Trpm5*-targeting shRNA were similar to those recorded from astrocytes transduced with scramble shRNA (black, n = 6 mice). This result is in line with the lack expression of *Trpm5* in astrocytes as reported in the optic nerve (Choi et al., 2015). These new data are now illustrated (Fig. 2h and 2k), added in the result section (page 7, line 22 to page 8 line 4) and discussed (page 18, lines 3-8). The method section has been updated regarding the recording condition of astrocytes (page 20 lines 29-34) and the immunohistochemistry against GFAP (page 22 lines 16-17).

More regarding shRNA knockdown: there is no quantification of the attenuation. There are a number of mechanisms to quantify the knockdown: Western blot for *Trpm5* protein, quantitative PCR or multiplex in situ hybridization to measure *Trpm5* transcripts quantitatively in AAV-shRNA-transduced vs. non-transduced motoneurons. One should quantify and report how much the shRNA actually knocked down the transcripts or attenuated protein expression. GFP expression only reports that the viral payload was dropped off and at least some of the payload was expressed.

We agree with the reviewer. This important point has also been raised by the reviewer 3. To address it, we performed additional experiments by using different techniques to quantify the *Trpm5* attenuation after shRNA injection. First, we analysed in culture cells the knockdown

efficiency of the shRNA-Trpm5 on mRNA level by quantitative real-time PCR. We demonstrated strong decreases (~95%) of the level expression of Trpm5 transcripts 48 h after HEK cells were transfected by a plasmid carrying a Trpm5-shRNA construct. ShRNA-Scramble had no effects. Second, by means of RT-PCR of the lumbar spinal cord we quantified a decrease of ~50% of Trpm5 mRNA from mice injected with ShRNA-Trpm5 versus ShRNA-Scramble. Third, by means of Western blots of lumbar spinal cords we observed that the decrease of ShRNA-Trpm5 was accompanied by a significant decrease of the membrane protein expression of Trpm5 by ~ 15%. Finally, patch-clamp recordings showed a decrease of ~60% of Trpm5-mediated I_{CAN} current amplitude from mice injected with ShRNA-Trpm5 versus ShRNA-Scramble. Altogether, these new data validate the Trpm5 knockdown. The new data are now added in the manuscript (page 7 lines 8-15) and illustrated (Fig. 2e and 2f). Protocols used for HEK cultures, RT-PCR and Western blots are added in the Methods section (page 22, line 25 to page 23, line 24).

One cannot compare the amplitude of ventral nerve output across Trpm5^{+/+} and Trpm5^{-/-} mice. That occurs at line 263 with reference to Supplementary Fig. 7c. The comparison is misleading because suction electrodes can generate vastly different signals depending on the quality of the seal on the ventral root. The measurements themselves have no biophysical meaning (unlike intracellular recording). So, the amplitude can vary between different preparations including the root and electrode diameter (how tight is the fit?) as well as whether the glass is fresh or the electrode is being reused after cleaning. I am not convinced the measurements in Supp. Fig. 7c are meaningful or reliable. In contrast I trust the before/after comparison in Supp. Fig. 7b because TPPO is being applied to the same preparation and thus one can be confident the seal is the same before and after the drug is applied.

We agree with the reviewer. We thus removed the amplitude data from the Supp. Fig. 7b

There are major developmental irregularities in the data, which are not explained. Consider the rotarod tests described in lines 276-283. Fig. 7d shows data from 4-week-old mice, in which the Trpm5^{-/-} mice severely underperform compared to wild type. In contrast, Supp. Fig. 8g shows data from 5-week-old mice in which Trpm5^{-/-} and wild type mice perform equally well. That is extraordinary compensation in 1 week. The locomotion in Trpm5^{-/-} mice, and Trpm5-Trpm4 double knockout mice, is intact and matching at 3 weeks of age, thus I wonder about the base of support and righting reflex at 5 weeks and later adult stages? It seems that the later in development the animals are studied, the less phenotype they exhibit. Yet the only test measured as far out as 5 weeks is rotarod, and the Trpm5^{-/-} mice are fine. The authors should also measure base of support and righting reflex out to 5 weeks too and determine whether the righting reflex and base of support observations normalize at 5 weeks like rotarod performance. Perhaps all the phenotypes will be equally well compensated in the Trpm5^{-/-} mice or the Trpm5-Trpm4 double knockout mice by 5 weeks. If so, it could be that Trpm5 is only important in early postnatal development and has no real relevance in the adult. That developmental story could be interesting.

We performed additional experiments and we observed that the increased base of support from neonatal Trpm5^{-/-} mice persists in young adult (3-4 weeks old) but is no longer present after 5 weeks of age. These new data are added in the Figure 7b.

We could not measure the surface righting reflex from adult mice because this test is not adapted for adult measurements. Indeed at this age, the animal places on its back rights itself in mid-air before the grip is released.

It appears that motor deficits in *Trpm5*^{-/-} mice (Fig. 7a-g) fade with age to end up being comparable with wild-type mice. We agree with the reviewer that apart from a compensatory mechanism that is usual in KO mice, we cannot exclude a transient role for *Trpm5* in the motor development, although in this case an improvement in motor performance from the ShRNA-*Trpm5* mice would be expected. This possibility is now clearly mentioned in the discussion (page 17, line 20; page 18, line 3).

Regarding locomotion vs. swimming, the main take-home message of the paper is that *Trpm5* is important for postural control. Postural muscles must function during locomotion but there is no locomotor phenotype up to 3 weeks of age. However, the animals have an obvious swimming phenotype from P5-P12, shortly before 3 weeks of age. Postural control is not relevant to swimming, where the bodyweight is mostly supported by the buoyancy in water. It is hard to reconcile why there is a swimming phenotype -- but not a locomotor phenotype -- in *Trpm5*^{-/-} animals if the hypothesis that *Trpm5* is important for posture is correct.

There is likely a misunderstanding. *Trpm5*^{-/-} mice displayed a locomotor phenotype because they are walking with a wider base of support depicting a postural deficit. We rewrote results to make this point more clear and to avoid any confusion (page 12, lines 23-25). However, the role of *Trpm5* is not limited to a postural control. We argue that *Trpm5* also plays an important role in amplifying the locomotor outputs notably required for high-demanding locomotor tasks. The evidence is that during a swimming activity for which postural control is less relevant, genetic deletion of *Trpm5* impairs swimming ability in young animals especially as fur-related buoyancy is less important compared to adults. In line with this interpretation, juvenile *Trpm5*^{-/-} mice failed to adapt locomotion to accelerated speed in the rotarod test, while during unforced locomotion they walked quite normally. Finally, we showed that the ability of the spinal locomotor network to amplify motor outputs is reduced in the absence of *Trpm5*. Altogether, these results suggest that *Trpm5* is recruited during locomotor tasks that require greater force such as when locomotion shifts to a higher speed.

We became aware that the multiple behavioural roles of *Trpm5* were not clear enough in the previous version of our manuscript. We thus rewrote several parts of the manuscript to correct this. First, to emphasize less on the postural role of *Trpm5* we change the title of the manuscript by replacing "postural" by "motor". At the end of the abstract and introduction it is now specified that our data suggest a dual role of *Trpm5* in both posture and locomotion. Finally, we rewrote some parts of the discussion to make clearer the task-dependent role of *Trpm5* (page 17, lines 6-14; page 18, lines 12-15).

The righting reflex data from *Trpm5*^{-/-} mice (Fig. 7a) and *Trpm5*-shRNA mice (Fig. 7f) are wildly variable. The righting reflex varies between ~3 sec to 120 sec in the experimental groups. That massive variability should be addressed. The control groups vary a lot too. Perhaps the authors should justify why this is still a good test when the animals vary so much. I have a hard time seeing signal through the noise. While I recognize one can always compute a mean but with this much variability I wonder if the data violate the assumptions of the central limit theorem, which precludes parametric statistics. The Methods specify the use of parametric and non-parametric tests, but in Figs. 7a & f it is not clear what test was applied, however I assume it would be an ANOVA for multiple levels of the independent variable.

As depicted by Altman and Sudarshan in 1975 (Anim Behav) a strong motor variability from pup to pup (including the righting reflex) is usual during the neonatal period and decreased with age as observed in our study. To support this point, we performed additional experiments and we confirmed this variability. This latter is coherent with the fact that a

number of researchers argue that variability in the early movements is an essential aspect of normal motor development (Thelen, 1995, Developmental psychobiology). Notably the righting reflex test is a neurodevelopmental test which is appropriate for evaluating the development of motor skills in neonates (Nguyen et al. 2017; Brooks et al. 2009). As suggested by the reviewer, we applied a two-way ANOVA with Sidak's multiple comparisons test (see methods and figure legend).

The developmental stage of the *in vitro* cellular electrophysiology is unclear. The methods do not specify the age of the animals used, just "neonatal and young adult" but that is quite vague. Especially given the major changes in the righting reflex that occurs from P1 to P12, that behavior changes dramatically. One assumes that the motoneurons are also changing a lot during that developmental time period. It is essential that the authors specify the postnatal day of the preparations for cellular studies.

To address this point, we clarified all the ages of mice on which we performed the different experiments. See methods (page 19, 1st paragraph).

Furthermore, the reader needs to know whether the properties measured so elegantly in Figures 1-4 hold steady during the entire range throughout which the behaviors sometimes change. Given the major performance improvement in rotarod at 5-weeks, I would like to see the motoneuron properties from 5-week-old slices (adult slice technologies are quite good now). The authors must document whether bistability and Trpm5 generate the same or similar properties in motoneurons in adult stages older than 5 weeks by which time the Trpm5^{-/-} knockout mice no longer have a phenotype in the rotarod test.

In vitro recordings from adult spinal motoneurons remain extremely challenging for recording large (> 400 μm^2) motoneurons, our cells of interest. To answer the question, several unsuccessful attempts have been made from 5-week-old slices. This is likely due to the fact that large motoneurons are the most sensitive to ischemia and to mechanical damage of their dendritic arborisation making difficult if not impossible to record bistable motoneurons in adult slice preparations. No previous studies have recorded bistable motoneurons from adult slice preparations in mammals although as mentioned in the introduction several evidences show that the plateau potential is part of the physiological repertoire of spinal motoneurons in mammals. When we discussed the potential transitory developmental role of Trpm5 in bistability we now stipulated that "further study will be required in the future to specifically investigate the role of *Trpm5* in bistable properties of motoneurons in adults" (page 18, lines 1-3). For that, *in vivo* recordings of lumbar motoneurons seems to be the best option. In this perspective, the acquisition of this expertise within the team or a collaborative work will be necessary.

It is unclear why (as reported on lines 293-295 and Fig. 7f) Trpm5-shRNA mice have such a severe locomotor deficit that they cannot even be tested on the rotarod after the 3rd postnatal week. As recapped above, Trpm5^{-/-} mice perform equally to wild-type mice by 5-weeks of age. So, why is there such a severe phenotype in the shRNA Trpm5 knockdown? In general, shRNA knocks down gene expression by ~50%, whereas a gene knockout is complete (100%). So why should the phenotype in the Trpm5-knockdown be much more severe than in the Trpm5-knockout? It doesn't make sense unless we consider that germline knockout transforms the brain and CNS in which case we have to question all the *in vitro* data in Figs. 1-4 because those motoneurons would have been transformed through germline knockout.

We understand the astonishment of the reviewer we share. At this stage, we can only provide intelligible elements for a discussion. As mentioned in the manuscript we posit that

this difference finds its origin in a compensatory mechanism present in the mutant mice that is ultimately less or not present in ShRNA-Trpm5 mice. The chronic lack of Trpm5 from the conception in mutant mice likely allows a stronger compensation beginning from the earliest embryonic developmental stages. Considering the time scale for transducing the *Trpm5*-targeting shRNA into motoneurons from mice intrathecally injected at birth with an AAV, the knock-down of Trpm5 in mice will inevitably occur much more later in the development, in a phase considered as a critical period in motor development (Walton et al., Neuroscience, 1992). This may contribute to a lack of compensation phenomenon which will lead to irreversible motor deficits.

We also observed that the proportion of large motoneurons able to self-sustained spiking is less important in ShRNA-Trpm5 mice (~30% of MNs) than in Trpm5^{-/-} mice (~50% MNs) (Figure 3c and 3d). In mice models of ALS disease, ~35% of large motor units lose their ability to discharge repetitively in the presymptomatic phase of the disease. This proportion increases to ~70% when ALS mice present severe motor deficits (Martínez-Silva et al. 2018, Elife). Therefore, in our study, the difference in the proportion of large bistable motoneurons may also contribute to the switch in the behavioral phenotype from mild to severe motor deficits.

In general, germline knockout mice are crude tools because (in this case) the gene is removed from the whole organism (Trpm5 is expressed in more than the nervous system). It would have been advantageous to limit the knockout to motoneurons using a Chat-cre mouse crossed with a mouse featuring a LoxP-Stop-LoxP (LSL) floxed-allele of Trpm5 to limit the Trpm5 knockout to cholinergic motoneurons.

We thank the reviewer for this interesting suggestion. Unfortunately, such approach would not prevent the knock-out of Trpm5 in all cholinergic cells from the whole organism making it difficult to interpret the role of Trpm5 from motoneurons in motor behaviours, insofar as spinal cholinergic V0c interneurons play a significant role in regulating locomotor outputs (Zagoraïou et al., 2009, Neuron). We favoured an intrathecal injection of AAVs for knocking down Trpm5 channels. Fortunately, the AAV expression was mainly restricted to MNs likely due to the uptake of AAVs by unmyelinated ventral roots in neonates, and did not affect other neurons with the exception of some astrocytes (see Figure 2h). Importantly, electrophysiological properties of astrocytes are not altered in ShRNA Trpm5 compared to ShRNA control (see Figure 2k). This result is in line with the lack expression of Trpm5 in astrocytes as reported in the optic nerve (Choi et al., 2015).

In Figure 5 and the text the authors describe a multi-compartment NEURON model which they use to test whether a Trpm5 mediated conductance can underly the I_{CaN} current and reproduce the observed bistability and self-sustained firing in motor neurons. The overall results presented in Figure 5 are confirmatory but do not add any novel insights that could be subsequently tested in their in vitro experiments.

We thank the Reviewer for estimating our modelling results as confirmatory and for the constructive critics. To meet the above concern, we performed new computational experiments, in which modelled motoneurons were stimulated synaptically (see above the response to Reviewer #1, who expressed similar concern). With this new simulation protocol, we obtained data (new figs. 5g-l) that were not available in the present biological experiments. In the latter, periodical stimulation of dorsal roots obviously led to mono- and polysynaptic excitation of motoneurons via synapses of different unknown location. To overcome restrictions of those biological experiments and check the robustness of the bistable behaviour observed in electrically stimulated model (new figs. 5d-f) we now used sinusoidal changes in synaptic conductances as physiologically relevant inputs instead of

periodic current pulses. Such synaptic excitation was addressed either to the soma or to three dendritic branches (see above) that had dual effect: voltaic (change in the membrane potential) and resistive (change in the membrane conductance contributed by synaptic channels).

Figure 8 is unnecessary as it simply restates the interpretation of the results found on page 16, without adding anything further. It would be more suitable to rework Figure 8 into a graphical abstract.

To address this point, we simplified Figure 8.

MINOR (but still important) CONCERNS

Line 95: What is "large" in this context? Large compared to what? Do the authors have some cell dimensions for alpha motoneurons vs. (for example) Ia interneurons in an adjacent lamina?

We mentioned it in methods page 20, lines 22-23 "from L4-L5 motoneurons with the largest soma ($> 400\mu\text{m}^2$) located in the lateral ventral horn".

Line 97: I do not agree that TEA at 10 mM will block "most" of the K⁺ channels. That concentration of TEA is on the low side. Yes, it will block some forms of K⁺ channels, but it will be insufficient for many others to be fully blocked. Therefore, I recommend that the authors instead write "Namely, after attenuating voltage-gated Na⁺ and K⁺ channels..." That rephrasing will be more accurate and does not impact how the authors interpret their data at all.

Done

Lines 98-105: I think the authors should state that the spikes in Figure 1 in the presence of TTX/TEA are Ca²⁺ spikes and perhaps even include an inset to show the properties of a Ca²⁺ spike in TTX/TEA vs. a Na⁺ spike in control aCSF using a faster sweep speed so the reader could see the difference.

Done. See Supplementary Figure 1

Line 107: Fig. 1h appears to emphasize temperature dependence of the plateau but not so much on brief supra maximal depolarization.

As illustrated in the Fig. 1g, the brief supra maximal depolarization was not affected by the temperature.

Line 112: Supplementary Fig. 1c is not much help because the motoneuron was tonically active to begin with. One cannot evoke a plateau or self-sustained firing from a tonic state.

To clarify the point we significantly modified Supplementary Fig1.

Line 166: "Ins(1,4,5)P3" should be formally defined. After writing it out they should define "IP3" as the abbreviation throughout (e.g., Line 167). However, the authors also use "InsP3" (Line 169). Please use only one abbreviation.

Done.

Lines 203-204: Please add the raw data points to Figs. 4c and 4d. All the other panes show the raw data points in addition to mean and standard deviation.

Done.

Line 207: Add raw data points for Supplementary Fig. 6c.

Done.

Line 227: What does "off" mean? Is G_TRPM5 set to zero?

Yes. To avoid any confusion we replace "off" by "set to zero".

Reviewer #3 (Remarks to the Author):

The paper submitted by Bos et al builds on the group's prior paper (Bouhadfane et al 2013) which showed that bistability of motor neurons in neonatal rat is calcium-activated, sodium-mediated, and temperature sensitive, and suggested that the currents underlying bistability were mediated by TRPV2 channels. Here, they demonstrate that Trpm5 channels are the main carriers of the I_{CaN} which is important for mediating motor neuron bistability using pharmacology, knockout mice, and shRNA-mediated knockdown. Further, they go on to describe the mechanism of activation and inactivation of the Trpm5 by pharmacologically targeting potential regulators of intracellular calcium to determine which affect the slow afterdepolarization. Finally, genetic/shRNA-based silencing of Trpm5 channels is shown to lead to deficits in hindlimbs.

The findings will be of broad interest to the field. The L-type calcium channel has long been implicated in motor neuron bistability. Where this work does not directly contradict that, it suggests that the L-type calcium channel plays more of a supporting role, rather than directly underlying the slow depolarizing current. The study is comprehensive in terms of investigating mechanism of the sADP in detail. The experiments are well-considered and well-executed. Additionally, the manuscript is well-written and logically laid out. Overall, the figures are clear. It is an exciting study but there are some issues to be addressed:

1. The age of the animals and preparations should be made clear in every section. This is particularly the case for the motor neuron recordings. The Methods states "neonatal and young adult" (line 420). That is a large range and young adult is vague. Prior work of the group has shown differences in motor neuron properties within the first week of age (in rat). Additionally, the expression levels of L-type calcium channels are low in neonates.

We agree. As requested, the age range is now clarified for each group of experiments in methos section (page 20, 1st paragraph).

2. The ΔV is used as a measure of "bistable ability" (line 108). The way in which this is calculated is not clear. For example, V_{th} is defined differently in the text and the legend. Is V_{th} the "most depolarized holding potential at which self-sustained firing can be triggered" (line 110) or the "spiking threshold" (i.e. line 1047, Supplementary line 10)? If the first, what does "th" stand for? If the second, is it the voltage threshold for action potential generation or the voltage of the cell at the rheobase step? The figure referred to when it is first defined (Supp Fig 1) is too condensed to be able to determine this. It is clear from the text (in several places) that V_{pp} is the most hyperpolarized voltage from which bistable firing can be triggered. However, the "pp" in V_{pp} is not obvious nor defined. Additional details as to how it is measured should be described (either in Results or Methods). Is it determined by 25pA steps? I would expect the V_{pp} to change based on current step amplitude and duration as

well but at least the amplitude of the current step seems to vary, judging by the figures. What parameters are constant for comparison between cells?

To address this important point, we have now modified Suppl Fig. 1 and corrected/rephrased methods section for clarifying the measurements (page 31, lines 4-8). We also redefined delta V in results section (page 5, lines 17-20). As pointed out by the reviewer, there is a mistake in methods. The ΔV measure did not consider the spiking threshold but the most depolarized holding potential (close but below the spiking threshold) at which the motoneuron can still generate a self-sustained spiking. We now defined the most hyperpolarized and the most depolarized holding potential at which the motoneuron generate a self-sustained spiking as " V_h min" (rather than V_{pp} in the previous version) and " V_h max" (rather than V_{th} in the previous version), respectively. As already mentioned in methods, we always used 25pA steps for calculating delta V (page 30, lines 32-33) "For 2s before injecting the depolarizing pulse, incrementing holding currents (steps of 25 pA) were injected into the cell until reaching the spiking threshold."

3. In Fig 2e and f, the AAV-GFP expression looks to be exclusively in the motor neurons. The promoters (CMV or U6) are not targeting motor neurons. How widespread is the GFP (or shRNA) expression? If it is restricted to motor neurons, how is that the case? In lines 153-154, how many cords is the quantification of labeled motor neurons from and what is the variance cord to cord? For the recordings related to Figure 2, this information is not critical (because GFP+ motor neurons are being targeted) but this is important for whole cord experiments and behavioral outcomes in Figs 6-7. For example, how are effects seen in the top half of Fig 6 related to motor neurons exclusively?

See the answer of the 2nd major concern of the reviewer 2.

4. The effects seen with the shRNA are dramatic so it is clearly having an effect but there is no validation of Trpm5 knockdown shown. A quantitative measure of the degree to which the channel is knocked down would be ideal.

See the answer of the 3rd major concern of the reviewer 2.

Also, due to the strong effects, particularly in the 3 week old mice in experiments related to Fig 7, are motor neurons still present/alive?

There is likely a misunderstanding. Behavioral tests in ShRNA-Trpm5 mice were performed until 12-days old age. At this age motor deficits were already strong, without observing any morphological or electrophysiological abnormalities (with the exception of bistable properties) from our immunohistochemical experiments (Fig 2h) and intracellular recordings (Supplementary Table 1), respectively. We are thus confident that lumbar motoneurons transduced by the Trpm5-shRNA are still present and alive and therefore motor deficits are not related with motoneuron death.

5. It is not clear why (or how) the data in Fig 7b, 7c, 7e, and 7h (and Supp Fig 8a, 8e, and 8f) are being normalized. The controls are not the same mice and individual data points are shown. Displaying the raw values for width, index, distance/set time, velocity, etc. on the y-axes would be more informative.

To address this point, we made the requested changes and we plotted all raw values. Note that the regularity index is still expressed in percentage because it represents the percentage of regular step pattern.

Minor

1. Please clarify if recordings were from lateral motor neurons or lateral and medial motor neurons.

This point has already been specified in results (page 5, lines 4-5) “We first ascertained that in mice, large ventrolateral motoneurons recorded in L4-L5 (Fig. 1a).” and in methods page 20, lines 21-22 “*In vitro* recordings and stimulations. For the slice preparation, whole-cell patch-clamp recordings were performed using a Multiclamp 700B amplifier (Molecular Devices) from L4-L5 motoneurons with the largest soma ($>400\mu\text{m}^2$) located in the lateral ventral horn.”

2. It might be useful to see at least one set of Ca^{2+} and Na^{+} spikes on an expanded time scale. It was impossible to tell what Vth on Supp Fig 1 was referring to and one cannot see the differences in Fig 1f and g.

To address this point, we plotted Na^{+} and Ca^{2+} spikes in Suppl. Fig.1

3. In addition to the reported n for the number of cells, please state the number of mice those cells came from, particularly for the experiments involving low numbers of recorded cells.

To address this point, we mentioned the number of mice the cells came from in all figure legends.

4. I suggest considering switching Figures 3 and 4 (and associated text). I think it would make for a more logical progression as Figures 2 and 4 are closely related.

Done

5. Supplemental fig 3b: Means for instantaneous firing frequency do not appear to match the data. (It looks like a figure editing error since means are identical to panel a.)

Corrected

6. Lines 281-3: This should be framed as a hypothesis as there are other possibilities (i.e. the channels are not important/relevant in adults).

This point is now clearly mentioned in the discussion (page 17, line 25).

7. Lines 411: I suggest rephrasing to be more similar to the last line of the Abstract, rather than “postural muscle activity”.

Done

8. The manuscript was very well written overall. There were just a few word choice errors and typos.

- Line 48: remove “downstream” Done

- line 49: spell out SERCA for first use Done

- line 50: remove “of” Done
- line 60: “in” should be “of” Done
- line 61: remove binary? Done
- line 75: “be” should be “being” Done
- line 106: “nearby” should be “near” Done
- line 144: “on” should be “to” Done
- line 145: “attested” to “confirmed”? Done
- line 206: should be “and were similar” Done
- line 274: “to walk” should be “from walking” Done
- line 286: “showed” should be “shown” (but showed is correct later in that same paragraph) Done
- line 332: “highest” should be “higher” Done
- line 341: should be “Furthermore, 2-APB directly inhibits.....but does not decrease....” Done
- line 479: intensity range? Mentioned
- line 760: the “s” in sADP is slow, not small, right? Corrected
- line 776: “whether” should be “when” Done
- line 1034 and Supplemental lines 5-6: supra- and subliminal should be suprathreshold and subthreshold Done
- line 1067 and 1095: “irrelevant” should be “scramble” (if that is what it was) or “control”? Done
- Fig 1c x-axis: “no. spikes” or “# spikes” Done
- Fig 3h (right): y-axis should be area Corrected
- Fig 5e (bottom 2): y-axis should be mean Corrected
- Supp Fig 5c y-axis: ohms symbol was lost Corrected
- Supp Fig 6b: labels should be changed for clarity, perhaps “remained bistable in drug” and “no longer bistable in drug”? Done

REVIEWERS' COMMENTS

Reviewer #1 (Remarks to the Author):

The authors have done an excellent job of revising this manuscript and responding to my comments. It continues to be an important new study, redefining our understanding of the persistent inward currents that are the basis of much of normal motor output

Reviewer #3 (Remarks to the Author):

I am satisfied with the authors' thorough responses and the related improvements made to the manuscript. I congratulate the authors on their study.